# Revisiting Visual Model Robustness: A Frequency Long-Tailed Distribution View

**Zhiyu Lin**[1], **Yifei Gao**[1], **Yunfan Yang**[1], **Jitao Sang**[1,2*]
[1]Beijing Jiaotong University, China
[2]Peng Cheng Lab, Shenzhen 518066, China
{zyllin, yf-gao, yfyang, jtsang}@bjtu.edu.cn

## Abstract

A widely discussed hypothesis regarding the cause of visual models' lack of robustness is that they can exploit human-imperceptible high-frequency components (HFC) in images, which in turn leads to model vulnerabilities, such as the adversarial examples. However, (1) inconsistent findings regarding the validation of this hypothesis reflect in a limited understanding of HFC, and (2) solutions inspired by the hypothesis tend to involve a robustness-accuracy trade-off and leaning towards suppressing the model's learning on HFC. In this paper, inspired by the long-tailed characteristic observed in frequency spectrum, we first formally define the HFC from long-tailed perspective and then revisit the relationship between HFC and model robustness. In the frequency long-tailed scenario, experimental results on common datasets and various network structures consistently indicate that models in standard training exhibit high sensitivity to HFC. We investigate the reason of the sensitivity, which reflects in model's under-fitting behavior on HFC. Furthermore, the cause of the model's under-fitting behavior is attributed to the limited information content in HFC. Based on these findings, we propose a **Ba**lance **S**pectrum **S**ampling (BaSS) strategy, which effectively counteracts the long-tailed effect and enhances the model's learning on HFC. Extensive experimental results demonstrate that our method achieves a substantially better robustness-accuracy trade-off when combined with existing defense methods, while also indicating the potential of encouraging HFC learning in improving model performance.

## 1 Introduction

The disparity between deep visual models and the human visual system (HVS) in image signal processing enables the models to outperform the HVS on vision benchmarks [16, 33]. However, it is worth noting that most visual models lack the robustness exhibited by HVS [36, 17]. Specifically, the generalization advancement of visual models can be partially explained by the utilization of human-imperceptible information, such as high frequency components (HFC) [4, 39, 29]. Additionally, the adversarial vulnerability is also closely related to this information, as observed in adversarial examples phenomena [36]. The coexistence of both conditions gives rise to a widely discussed hypothesis: *the vulnerability of the model stems from the utilization of imperceptible high-frequency information in images*.

Based on this hypothesis, a line of research has been conducted to explain and enhance the robustness of the model [43, 1, 7, 9, 8, 35, 13]. However, these works share a common limitation: most of them solely rely on qualitative analysis of frequency components observed by humans, without further delving into the statistical properties. This limitation gives rise to the following issues in existing

---

[*]Corresponding Author

37th Conference on Neural Information Processing Systems (NeurIPS 2023).

works: (i) the division of the frequency spectrum is intuitively based on manually set radius, resulting in inconsistent conclusions across different datasets and models; and (ii) The understanding of high frequency utilization is limited. For example, improving model robustness often involves suppressing the learning on HFC, which is in stark contrast to the promotion of high frequency utilization to improve generalization performance. This leads to a trade-off between robustness and accuracy in some defense methods.

In this paper, to overcome the challenges posed by the aforementioned limitation, we are motivated by the long-tailed characteristics in the spectrum. Specifically, the power spectral density of an image naturally follows a power-law distribution, where a few low-frequency bands accounts for the majority of the spectrum power while a large number of high-frequency bands maintains a low power distribution. This new realization encourages us to revisit the significance of high-frequency components in addressing the problem of model robustness under the guidance of deep long-tailed learning.

In deep long-tailed learning [46], the presence of a large number of tail classes plays a critical role in improving model performance. These tail classes offer benefits such as ensuring data integrity and diversity [5] and capturing rare events [28]. However, it is commonly observed that models tend to under-fit these classes due to their numerical disadvantage. Intuitively, the long-tailed characteristics of the frequency spectrum also exhibit similar phenomena. We confirm the model's (standard training) under-fitting behavior on high frequencies through the lens of loss landscape [27]. Under the context of model under-fitting, the discussion regarding HFC and model generalization performance can be compatible in most studies [29, 39], as the common underlying cause is the low generalization resulting from under-fitting. In this paper, we establish a link between under-fitting behavior and model robustness, specifically identifying that the model's under-fitting of high-frequency components leads to high sensitivity towards them. We validate this relationship across different datasets and models, consistently confirming our findings. Model's under-fitting behavior on high frequency leads to both low generalization and high sensitivity towards HFC. Thus, it is crucial to improve the under-fitting behavior and further mitigate the trade-off between accuracy and robustness.

Drawing inspiration from deep long-tail learning, we introduce spectral entropy as a measure of the information content of frequency components and provide an explanation for model's under-fitting behavior on HFC. We attribute model's under-fitting behavior to the low spectral entropy of HFC. Based on this perspective, the principle of maximum entropy [23, 3, 32] guides us in constructing an image spectrum with a more balanced energy distribution to address the issue of low information entropy. Finally, our quantitative experiments and theoretical analyses inspire us to propose a simple yet highly effective **Ba**lance **S**pectrum **S**ampling (BaSS) strategy, which can work in conjunction with different training paradigms and enables the model to achieve a better trade-off between robustness and accuracy.

Our major contributions are as follows: (1) To the best of our knowledge, we are the first to propose focusing on the long-tailed distribution in instance-wise , specifically in the frequency domain. This novel perspective provides insights for analyzing and improving model performance. (2) We revisit the relationship between HFC and model robustness, revealing the under-fitting phenomenon and its association with high sensitivity and low generalization at high frequencies. To address the trade-off between accuracy and robustness, we propose to enhance the high-frequency learning. (3) We explain the under-fitting behavior through the lens of spectral entropy. Drawing inspiration from the principle of maximum entropy, we propose a simple yet highly effective spectral sampling strategy (BaSS) to improve the model's under-fitting behavior. By incorporating BaSS with other training paradigms, such as adversarial training and AugMix, model achieves better trade-off between accuracy and robustness. Comprehensive experiments and analyses reveal the effectiveness of our method.

## 2 The long-Tailed Distribution in Frequency Domain

### 2.1 Notations and Preliminaries

**Image Processing in the Frequency Domain.** We perform 2-D discrete Fourier transform (referred to as the DFT) $\mathcal{F}$ on image $X \in \mathbb{R}^{H \times W}$ to get Fourier spectrum $\tilde{X} \in \mathbb{C}^{H \times W}$. Unless otherwise specified, the low frequencies are shifted towards the center of the Fourier spectrum. We denote $\tilde{X}[u, v]$ as the frequency component where $u \in \{0, \ldots, H-1\}$ and $v \in \{0, \ldots, W-1\}$. Addition-

ally, $\omega_{u,v}$ denotes the Fourier Basis function [38] and $r_{u,v}$ denotes the Euclidean Distance from the spectrum coordinate $[u, v]$ to the center of spectrum, where a larger distance correspond to a higher frequency, and vice versa.

**Power Spectral Density.** The power spectral density is defined by azimuthally averaging the magnitude of Fourier coefficients over radial frequencies. Specifically, the value of power density is estimated as the azimuthal integration (referred to as $AI$) over radial frequencies $\phi$ according to Eq.1.

$$AI(r_k) = \int_0^{2\pi} \|\tilde{X}[r_k \cdot \cos\phi, r_k \cdot \sin\phi]\|^2 d\phi \tag{1}$$

where $r_k$ is the Euclidean Distance from the $k^{th}$ frequency band to the center of the spectrum [6, 14], $k \in \{0, \ldots, H/2 + 1\}$.

## 2.2 Definition of Frequency Long-Tailed Scenario

Real-world data generally demonstrates a long-tailed distribution, where a small number of dominant categories comprise the majority of the data, and a large number of minority categories possess limited samples. Although previous studies [46, 37] have primarily investigated the degradation in model performance due to the long-tailed problem at the class level, they have not adequately considered the potential existence of a long-tailed issue within individual images. Through our analysis of the Fourier spectrum, we observe that the power spectral density follows a power-law distribution, which also exhibits the long-tailed characteristics. Specifically, the distribution (y-axis) of $i^{th}$ frequency band (x-axis) is calculated by the proportion of power in $i^{th}$ band to the total power of the spectrum, which is denoted as $\pi_i = AI(r_i)/\sum_k AI(r_k)$. Without loss of generality, $\{\pi_i\}$ naturally present in decreasing trend (*i.e.*, if $k_1 < k_2$, then $\pi_{k_1} < \pi_{k_2}$).

Consequently, we summarize the statistics of the frequency feature from the long-tailed model perspective as follows: (i) The head part, which is consisted of a few low-frequency bands, accounts for the majority of the spectrum power and is highly semantic and information dense; (ii) The tail part, which is composed of a large number of high-frequency bands, however maintains a low power distribution. This part includes high-frequency semantic information (e.g., texture) as well as human-imperceptible information.

$$LFC = \{\tilde{X}[u, v] \,|\, \max_n \sum_{i=0}^{n} \pi_i \leq 80\%, r_{u,v} \leq r_n\}$$

$$HFC = \{\tilde{X}[u, v] \,|\, \min_n \sum_{i=n}^{H/2+1} \pi_i \geq 20\%, r_{u,v} \geq r_n\} \tag{2}$$

**Long Tail based Spectrum Division.** Previous analysis [15, 39, 31] indicating that a few low-frequency components form the image's main content, while most high-frequency components comprise a minor part. The rule that few factors determine main outcomes is encapsulated by the Pareto Principle and the "80/20" quantitative rule, found across various domains, where $80\%$ of effects are caused by $20\%$ of the reasons. Guided by this rule, we defined the boundary between frequencies using an 80/20 spectral energy ratio. We formally define the low- and high-frequencies according to the Eq.2, where the $r_n$ is the Euclidean Distance from the $n^{th}$ frequency band to the center of the spectrum. The set composed of $\tilde{X}[u, v]$ occupying $80\%$ of the total power in the spectrum is defined as the low-frequency component, and the remaining portion of the spectrum is defined as the high-frequency component. We experimented with various datasets, finding that different resolution images' high-low frequency boundaries are at a $20\%$ radius from the spectrum's center (as shown in Appendix B.2). This confirms the stability and reasonableness of the classification system using spectral energy and the Pareto Principle.

## 3 Investigating and Explaining Model Robustness Behaviors: A Frequency Long-tailed View

Vision models have demonstrated a capability in learning frequency information [24, 39, 29, 43]. Intuitively, model's behavior could be affected by the long-tailed characteristics in the frequency

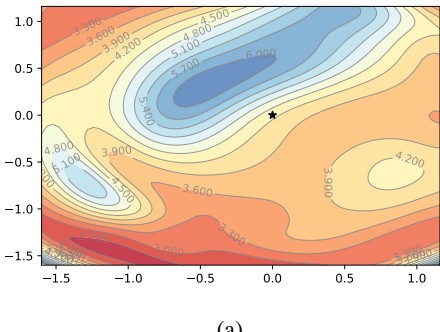 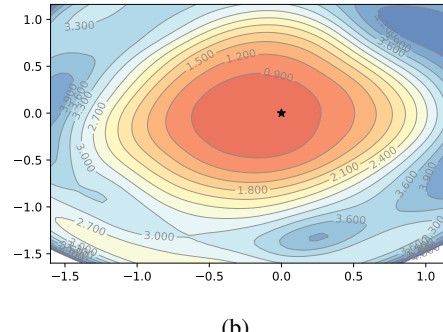

|    (a)    |    (b)    |

Figure 1: Loss landscape visualization of HFC (a) and LFC (b). We conduct experiments on ResNet-18 naturally trained on CIFAR10 dataset. The black star symbol represents model parameters $\theta$. A noticeable phenomenon is that the model exhibits a learning bias towards different frequency components: specifically, the learning for high-frequency components has not converged, while the learning for low-frequency components has already reached a local optimum.

domain, which has not been fully explored in previous studies, and is crucial for improving model performance in visual tasks. In this section, we first summarize the impact of long-tailed characteristics as a phenomenon of the model's under-fitting of HFC (Sec. 3.1). We then explore the model's sensitivity to HFC from the perspective of under-fitting, and revisit the relationship between HFC and robustness (Sec. 3.2). Finally we explain the model's under-fitting behavior from the perspective of spectral entropy (Sec. 3.3)

### 3.1 Model's Under-fitting Behavior on HFC

In the long-tailed theory, the tail part is considered more deserving of attention than the head part. On one hand, the tail part is often overlooked by the system due to its numerical disadvantage(e.g., the sample size of the tail class is far less than that of the head class in the context of class-wise long tail). On the other hand, it plays a crucial role in enhancing the system's performance, primarily due to two characteristics (i) providing data integrity and diversity [5], and (ii) capturing rare events [28]. In this subsection, we explore the learning behavior of models regarding frequency components. Specifically, we visualize and analyze the loss landscape [27] of the classifier on both LFC and HFC. We present the detailed experimental procedure in the Appendix C.

**Phenomenon of Model Under-fitting on HFC.** The convergence result of LFC and HFC learning exhibits significant differences as shown in Fig. 1. Fig. 1(b) suggests that the loss landscape on LFC exhibits a benign characteristic: the parameter space around the model $\theta$ is relatively flat and close to the local minima, indicating that the model has precisely converged on LFC. In contrast, the loss landscape on HFC shown in Fig. 1(a) manifests a dramatic non-convexity in certain regions. It can be observed that the model $\theta$ are located in a steep region of the landscape, with loss rapidly decreasing when moving along the gradient directions (i.e., which are perpendicular to the contours). Additionally, in the bottom-left region of the Fig. 1(a), we also observe a better local minimum far away from current model. These observations indicate that the model still has the potential to improve its ability to learn HFC.

**Positive Correlation between Generalization and HFC Learning.** It is widely recognized that under-fitting is a cause of low generalization performance. Recent works have confirmed that models exhibit lower generalization ability on HFC compared to LFC [39, 29]. Meanwhile, ignoring the HFC can lead to inaccurate representation of the data distribution, resulting in a degradation in model generalization [39, 4]. This correlation can be attributed to the first characteristic of tail components mentioned above. Hence, it is vital to devise and employ strategies that alleviate or eliminate the under-fitting of HFC, as it substantially contributes to the enhancement of model generalization. performance [39, 29].

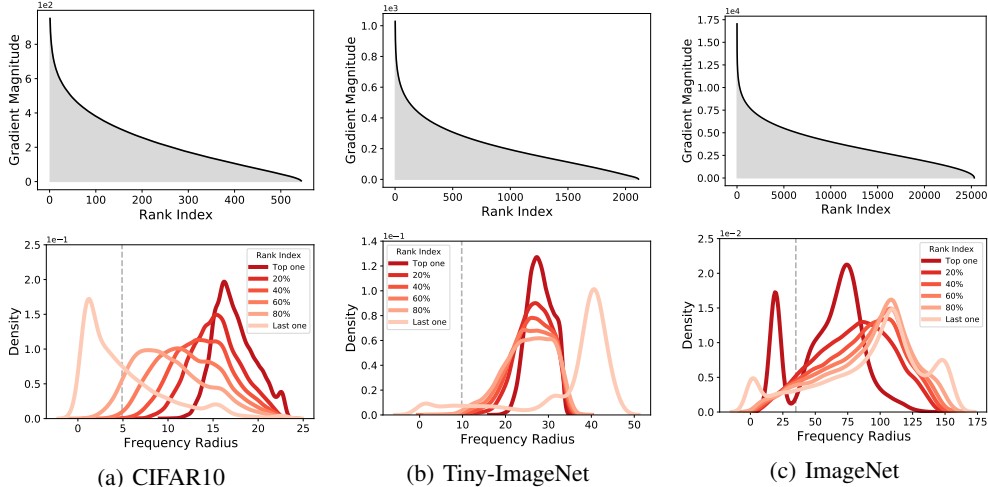

Figure 2: Model sensitivity to frequency components on CIFAR10, Tiny-ImageNet and ImageNet datasets. We present the sorted average gradient magnitude on test set in the first row, with the $x$-axis denoting the rank index. In the second row, we select part of rank positions, and display the density distribution of frequency radius. The gray dashed line is served as the boundary between low- and high-frequencies. The overall phenomenon is that naturally trained model is highly sensitive to tail frequency components.

## 3.2 Revisiting the Relationship between HFC and Model Robustness

### 3.2.1 Motivation

The phenomenon of under-fitting inspires us to revisit the issue of model robustness. Robustness is typically expressed as the sensitivity of model outputs to perturbations in inputs, and it can be evaluated in two ways: (i) changes of the outputs leaded by introducing additional perturbations to the inputs; and (ii) inputs gradients with respect to outputs. The results shown in Fig. 1(a) indicate that standard training classifier still exhibit significant loss values in HFC and may have large gradients with respect to high-frequencies, thus resulting in model vulnerability. In the context of long-tailed theory, the tail part usually represents special or extreme cases and thus is more likely to be affected by rare events, making HFC crucial for evaluating and analyzing model robustness. Existing studies [43, 7, 9] suggest that the vulnerability of models is attributed to HFC, however, this conclusion is not consistent across different datasets and models. We attribute the potential causes of inconsistency to the definition of HFC. We aim to further explore relationship between HFC and model robustness.

### 3.2.2 Analyzing Model Sensitivity to Frequency Components

Model sensitivity to different frequency components has been explored using the Fourier heatmap, which quantifies the changes in outputs when applying Fourier basis perturbations to images [43]. However, there are a few concerns about its applicability: (i) The theory of Fourier basis is based on linear networks while deep neural networks typically exhibit high non-linearity, (ii) The perturbations is excessively sensitive to the magnitude of the basis disturbance. Thus, we take inspiration from [29, 7] and employ the loss gradient across various frequency components to examine sensitivity.

**Estimation of Model Sensitivity.** Given an input image $X$ with label $y$, the output of classifier $f$ is $f(X)$. The objective function of model's training is formulated as $\mathcal{L}(f(X), y)$. Consequently, the gradient of the loss w.r.t. the original input $X$ can be expressed as $g = \nabla_X \mathcal{L}(f(X), y) \in \mathbb{R}^{H \times W}$, whose magnitude can indicate the sensitivity of the classifier to pixel domain perturbations. To further assess the model's sensitivity in frequency domain, we employ the set of orthogonal Fourier basis vectors $\{\omega_{u,v}\}$ and project $g$ into frequency domain coordinate system. The corresponding coordinate of projection in $\omega_{u,v}$ is defined as follows:

$$c_{u,v} = \langle g, \omega_{u,v} \rangle = \sum_{m=0}^{H-1} \sum_{n=0}^{W-1} g[m,n] \cdot exp\{-i2\pi \cdot (\frac{mu}{H} + \frac{nv}{W})\}. \tag{3}$$

The sensitivity of model to the frequency components corresponding to the spectrum point $[u, v]$ is measured by the gradient magnitude on $\omega_{u,v}$, which is defined as $\|c_{u,v}\|_2$. In order to identify the frequency domain sensitivity patterns of the model at the dataset level, we first sort $w_{u,v}$ according to $\|c_{u,v}\|_2$ on a single image. For ease of statistical analysis, we replace $w_{u,v}$ with the frequency radius $r_{u,v}$ as the feature vector. Finally, we get the set of feature vector for all images and estimate the density of frequency radius using kernel density estimation (KDE) within the same rank index. We analyze the frequency sensitivity across the CIFAR10 [25], Tiny-ImageNet [26], and ImageNet [12] datasets with resolutions of $32^2$, $64^2$, and $224^2$ pixels, respectively. We show detailed settings in the Appendix D.

**Results and Discussions.** From the first row of Fig. 2, we observe an imbalanced distribution of gradients in different frequency bands, and the degree of imbalance is quite pronounced, resembling a similar long-tailed distribution. The results in the second row of Fig. 2 indicate that the density peaks of frequency radius in high-magnitude gradients are mainly concentrated in the tail high-frequencies (i.e., represented by the darker colored curves). These observations clearly demonstrate the relationship between HFC and model vulnerability, indicating that perturbations in HFC are more easier to cause errors in the model. In the result of ImageNet as shown in Fig. 2, we notice a particular phenomenon where there are also large gradient magnitude in the head components. This could be due to the wide spectral range causing similar vulnerabilities in the head components. However, this phenomenon does not affect the consistency of our conclusions. It also suggests that addressing robustness in the ImageNet dataset is a more challenging task.

### 3.3 Reasons leads to the Under-fitting Behavior

In deep long-tailed learning, the lack of sufficient generalizable information on tail classes directly causes biased behavior in models. Similar phenomena inspire us to explain the under-fitting of the tail part from an information-theoretic perspective in the case of frequency long-tailed problems.

#### 3.3.1 The Spectral Entropy of HFC is Much Lower than LFC

**Spectral Entropy** To quantitatively characterize the amount of information contained in a given image, spectral entropy is utilized analogously to Shannon entropy in information theory. The continuous frequency domain is discretized and represented by a discrete variable $\xi$, with $\xi_l$ and $\xi_h$ as its minimum and maximum values, respectively. Given that the spectral density of natural images follows a power-law distribution, its density function can be computed and normalized as follows: $p(\xi = k) = \frac{1-\alpha}{\xi_h^{1-\alpha} - \xi_l^{1-\alpha}} \cdot k^{-\alpha}$, where $\alpha$ is the parameter determining the magnitude of the long-tailed effect, and its smaller value indicates a more significant long-tailed effect. We use $\pi_i$ to estimate $p_i$, and then the spectral entropy of $k^{th}$ band is derived by applying the Shannon entropy formula: $H_k = -\pi_k \log \pi_k$. Consequently, the flat spectrum corresponds to maximal spectral entropy, while the spectrum of a single frequency signal has minimal spectral entropy, which is zero.

**Theorem 1** *Given any natural image $X$, $k$ represents the order of frequency band, then the spectral entropy $H_k$ is given by: $H_k = (\alpha A_\alpha \log k + H(A(\alpha))) \cdot k^{-\alpha}$, where $A(\alpha) = (1 - \alpha) \cdot (\xi_h^{1-\alpha} - \xi_l^{1-\alpha})^{-1}$. Then we have, $\lim_{k \to 0} H_k = \infty$ and $\lim_{k \to \infty} H_k = 0$.*

Theorem 1 demonstrates an inverse proportional relationship between the frequency band and spectral entropy, decreasing at an $\alpha$ order. This indicates that high-frequency bands in the tail contain substantially less information than low-frequency bands in the head. Unlike measuring the number of image samples, we provide a new way to measure image information from the frequency domain perspective, explaining the model's under-learning behavior for low-information data. Moreover, Theorem 1 also suggests that high-resolution data, such as ImageNet, often lack effective generalizable information in the tail. This finding aligns with Fig. 2, illustrating that the model fails to generate gradient components in the ultra-high frequency range of ImageNet data.

#### 3.3.2 Balanced Spectral Entropy Benefits the Under-fitting Phenomena

The maximum entropy principle seeks to make reliable predictions on unknown data distributions by selecting probability distributions with the highest entropy, without introducing unnecessary biases or prior knowledge. Both generalization and robustness reflect a model's ability to infer unknown samples, which depends on its capacity to capture information during training. Based on

our analysis, we suggest that this principle explains the model's under-fitting on HFC: the model prioritizes utilizing LFC with higher information entropy to improve its capacity for fitting unknown data, leading to the observed under-fitting in high frequencies. To address under-fitting in HFC, we aim to increase the amount of information, and the maximum entropy principle offers a solution.

**Theorem 2** *Given the prior data $\xi$, and its corresponding probability distribution $p(\xi)$, the principle of maximum entropy consider the following optimization: $\hat{p} = \{arg\max_{p\in\mathbb{P}} \ H(p) \mid p(\xi) \geq 0, \int_{\Omega} p(\xi)d\xi = 1, \int_{\Omega} p(\xi)r_i(\xi)d\xi = \alpha_i, i = 1, \cdots m\}$. Without considering the moment constraints, the uniform distribution $p(\xi) = \frac{1}{\xi_h - \xi_l}1_{[\xi_l,\xi_h]}(\xi)$ satisfies the principle of maximum entropy.*

Theorem 2 demonstrates that a uniform distribution with balanced information adheres to the maximum entropy principle when fitting an unknown distribution. Intuitively, this corresponds to fair learning for tail classes, which has been extensively studied in deep long-tailed learning to improve tail class generalization using techniques like re-weighting or re-sampling. This insight drives us to tackle the under-fitting issue of HFC by redistributing spectral energy through a balanced sampling strategy.

## 4 Methodology and Experiments

### 4.1 Balanced Spectrum Sampling (BaSS)

Previous analyses encourage us to HFC learning by balancing the power spectral density. Nonetheless, it is essential to consider the potential negative effects of noise present in the image while implementing a balanced spectrum. Noise components, often found in ultra-high-frequency parts, have been proven less beneficial for classification tasks [39, 29]. Moreover, uniformly allocating noise and other frequency components may (i) amplify noise influence on model learning, and (ii) allocate less energy to remaining frequency bands, both negatively impacting model learning. To address these challenges and achieve spectral balance, we propose a spectral sampling strategy in this subsection.

$$\hat{\pi}_i = \frac{\log_\tau AI(r_i)}{\sum_{j=1}^{B} \log_\tau AI(r_j)} \tag{4}$$

**Sampling Strategy.** For the $i^{\text{th}}$ frequency band, we sample the power spectral density according to Eq. 4 with a probability of $\hat{\pi}_i$, where $\tau \geq 1$ denotes the base and $B$ is the maximum frequency band of the spectrum. Our approach is inspired by two background pieces of knowledge: solutions for the long-tail distribution of image categories and image sampling. The solution for the class-wise long-tail distribution adjusts the sampling frequency of samples with different class quantities through the inverse weighting of category quantities. However, features in images (like spatial pixels and spectral density) can not usually be directly chosen via the sampling theorem. Therefore, we refer to the method of sampling image pixels in the spatial domain: (i) discretizing image features that corresponds to the spectrum obtained from the Fourier transform; (ii) selecting and combining feature values from different regions: we re-weight features at different frequency bands based on the sampling probability $\hat{\pi}_i$ and obtain a new image through the inverse Fourier transform.

Specifically, we employ a logarithmic function to: (i) establish a smooth connection between the power density of head and tail parts, mitigating the impact of ultra-high frequency components on the overall distribution, and (ii) adjust spectral power values in the head part to a reasonable range, while diminishing the power disparity between head and tail frequency bands, yielding a more balanced distribution overall.

### 4.2 Working Mechanism Exploration of BaSS

In this subsection, we validate the potential of the Balanced Spectrum Sampling strategy (BaSS). Specifically, we examine two paradigms for utilizing the sampled images: (i) Data augmentation, where each training image is performed by BaSS with a probability of $\gamma$. In this paradigm, the model will be validated on a test set comprised only of natural images. (ii) New dataset, where both the training and testing sets consist entirely of sampled images.

**Experiments Settings.** We refer to the dataset performing BaSS on CIFAR10 [25] as CIFAR10-B. For different values of $\gamma$, we obtain a set of ResNet-18 [19] models with standard training. We

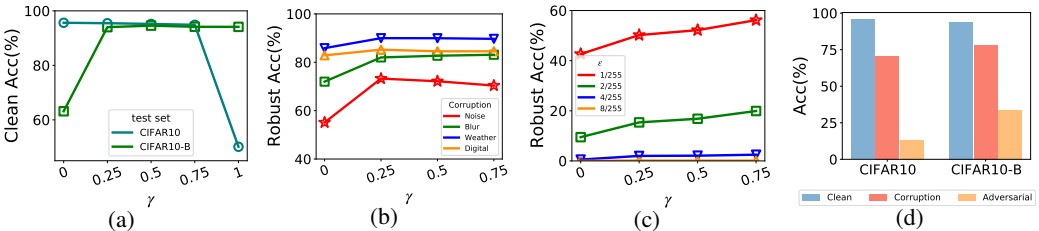

Figure 3: Results of exploring potention of BaSS. (a) clean data accuracy, (b) robust accuracy evaluated on CIFAR10-C, (c) robust accuracy against PGD attack with 20 steps, and (d) clean and robust acc comparison on CIFAR10 and CIFAR10-B

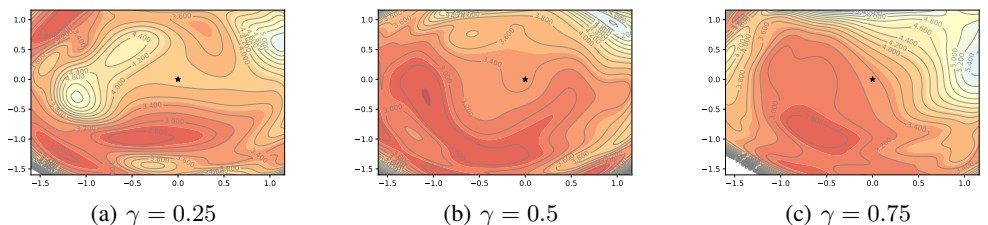

Figure 4: Visualization of Loss Landscape on models that training with CIFAR10-B as data augmentation at a ratio of $\gamma$. The black star symbol represents model parameters $\theta$.

use clean accuracy, corruption robustness and adversarial robustness as the metric to measure the potential of BaSS strategy. Specifically, (i) the clean accuracy is evaluated on both CIFAR10 and CIFAR10-B test set. (ii) To evaluate the corruption robustness, we first construct a corruption version for CIFAR10-B, referring to the construction method of CIFAR10-C [20] and denotes as CIFAR10-B-C. We then calculate the mean accuracy across 19 corruption types and 5 severities from CIFAR10-C and CIFAR10-B-C. (iii) We perform PGD attack [30] with 20 steps and $\ell_\infty$ constraint $\epsilon = 1/255, 2/255, 4/255, 8/255$ to evaluate adversarial robustness.

**CIFAR10-B as Data Augmentation.** The results in Fig. 4 indicate that as the proportion of CIFAR10-B images increases (from Fig. 4(a) to Fig. 4(c)), the model converges more on the high-frequency components (the black stars are closer to the local minimum). The results from Fig. 3(a), Fig.3(b) and Fig. 3(c) indicate that the data-augmented model achieves a better trade-off between robustness and accuracy. Specially, the corruption robustness is improved across all categories, further achieving a more balanced performance. The enhancement of adversarial robustness under small perturbation (e.g., $\epsilon$=1/255, 2/55) is quite significant, with the improvement magnitude positively correlated with the mixing ratio. However, the issue of model robustness under large perturbations still remains.

**CIFAR10-B as an New Dataset.** The naturally trained model achieves a better robustness-accuracy trade-off on the CIFAR10-B than on the CIFAR-10 dataset, as shown in Fig.3(d). What attracts us is the improvement in adversarial robustness. From the more detailed results in the Appendix F.2.1, the performance of CIFAR10-B is superior to that of CIFAR-10 and all mixed datasets. Based on the anaylsis shown in the Appendix F.2.1, we observed that the model trained on the CIFAR10-B dataset has a more balanced gradient distribution across different frequency domains. This indicates that when facing adversarial attacks, the model's vulnerabilities aren't concentrated in the high-frequency direction of the image. As a result, the adversarial attack process needs to generate adversarial noise containing patterns from multiple frequency domains. This insight suggests that we use the CIFAR10-B dataset in place of the original dataset during the adversarial training process.

### 4.3 BaSS Working in Conjunction with Adversarial Training

Adversarial Training has been the most successful defense strategy, where models are explicitly trained to be robust in the presence of adversarial attacks [30, 2, 34, 45]. In this subsection, We utilize the BaSS strategy to modify the input image and train the models in conjunction with PGD-AT [30], TRADES [45] and MART [40].

Table 1: **Clean and Robust Accuracy(%) on CIFAR10 and CIFAR100.**

| CIFAR10 | ResNet-18 | | | WRN-34 | | | CIFAR100 | ResNet-18 | | | WRN-34 | | |
|---|---|---|---|---|---|---|---|---|---|---|---|---|---|
| | Clean | $PGD^{20}$ | AA | Clean | $PGD^{20}$ | AA | | Clean | $PGD^{20}$ | AA | Clean | $PGD^{20}$ | AA |
| PGD-AT | 83.50 | 52.72 | 48.90 | 86.71 | 55.13 | 51.98 | PGD-AT | 56.33 | 29.29 | 24.88 | 61.78 | 30.48 | 27.93 |
| +BaSS | 89.22 | 59.61 | 55.90 | 91.08 | 63.65 | 60.91 | +BaSS | 63.91 | 31.61 | 27.95 | 67.07 | 34.87 | 32.21 |
| $\Delta$ | **+5.72** | **+6.89** | **+7.00** | **+4.37** | **+8.52** | **+8.93** | $\Delta$ | **+7.58** | **+2.32** | **+3.07** | **+5.29** | **+4.39** | **+4.28** |
| TRADES | 82.15 | 52.50 | 49.05 | 85.66 | 56.54 | 53.77 | TRADES | 58.30 | 29.90 | 25.52 | 61.06 | 31.83 | 27.09 |
| +BaSS | 87.20 | 60.01 | 56.27 | 91.20 | 65.38 | 62.69 | +BaSS | 64.17 | 32.94 | 27.77 | 69.01 | 37.03 | 33.03 |
| $\Delta$ | **+5.05** | **+7.51** | **+7.22** | **+5.54** | **+8.84** | **+8.92** | $\Delta$ | **+5.87** | **+3.04** | **+2.25** | **+7.95** | **+5.20** | **+5.94** |
| MART | 81.38 | 54.74 | 49.48 | 83.78 | 56.75 | 51.42 | MART | 57.80 | 30.73 | 26.26 | 58.10 | 33.46 | 28.36 |
| +BaSS | 88.75 | 60.43 | 56.93 | 89.63 | 63.71 | 59.09 | +BaSS | 63.94 | 32.95 | 28.01 | 64.02 | 37.79 | 32.20 |
| $\Delta$ | **+7.37** | **+5.69** | **+7.45** | **+5.85** | **+6.96** | **+7.67** | $\Delta$ | **+6.14** | **+2.22** | **+1.75** | **+5.92** | **+4.33** | **+3.84** |

Table 2: **State-of-the-art performance on CIFAR10 and CIFAR100.**

| Method | CIFAR10 | | CIFAR100 | |
|---|---|---|---|---|
| | Clean | AA | Clean | AA |
| DAJAT[2] | 88.71 | 58.04 | 68.75 | 31.85 |
| PORT[34] | 86.68 | 60.27 | 65.93 | 31.15 |
| **Ours** | **91.20** | **62.69** | **69.01** | **33.03** |

Table 3: **Clean and Robust Accuracy(%) on Restricted-ImageNet.**

| Method | Restricted-ImageNet | | | |
|---|---|---|---|---|
| | PGD-AT | +BaSS | TRADES | +BaSS |
| Clean | 65.13 | **66.76** | 64.79 | **66.90** |
| $PGD^{20}$ | 49.44 | **57.63** | 51.71 | **57.96** |
| AA | 33.37 | **41.77** | 35.50 | **42.80** |

**Experiment Settings:** We conducted experiments on the following three datasets: CIFAR10, CIFAR100 [25] and Restricted-ImageNet [12], using ResNet-18, WideResNet-34-10 [44] and ResNet-50 [19] as backbone. We compare the performance of our method with the baseline defenses PGD-AT [30], TRADES [45], MART [40], and also state-of-the-art defenses DAJAT [2] and PORT [34] from the Robust Bench leaderboard [10]. To fairly evaluate each model, we assessed (a) clean accuracy (i.e., clean test data), robust accuracy under (b) $PGD^{20}$ attack, the PGD attack [30] with 20 steps and (c) AutoAttack(AA) [11]. The $\ell_{\infty}$ attack perturbation was bounded to $\epsilon = 8/255$ with a step size of $2/255$ in both the training and test processes of CIFAR-10 and CIFAR-100. For Restricted-ImageNet, $\epsilon = 4/255$ with a step size of $1/255$. Detailed settings are available in the Appendix F.2.2.

**Results**. As shown in Tab. 1 and Tab. 2, by processing the dataset with BaSS strategy, our method consistently achieves a better robust-accuracy trade-off across datasets and architectures. Moreover, as shown in Tab. 3, our approach trained with WRN-34-10 achieves state-of-the-art performance compared with existing defenses on CIFAR10 and CIFAR100. Adversarial training has been observed to exhibit a more pronounced data hunger compared to natural training [2]. Existing defense methods that are capable of improving both generalization and robustness performance typically involve adversarial training on an enlarged training set [41, 18], which is accompanied by a substantial training overhead. It is worth noting that BaSS strategy barely increase the training cost (including the training time and the data size, comparison details are available in the Appendix F.2.2), and achieve a competitive performance. This further demonstrates the effectiveness of BaSS.

### 4.4 BaSS Working in Conjunction with AugMix

AugMix [21] generates complex and diverse images through data augmentation, improving the model's robustness to different types of corruption. Specifically, for the original data $x$, $x_{augmix}$ are generated by combining multiple simple data augmentation, then a consistency constraint is applied to the $x_{augmix}$ and $x$ during the training process. Experiments typically involve two AugMix processes to obtain $x_{augmix1}$ and $x_{augmix2}$. To verify that our method of increasing data information, BaSS, can be combined with the AugMix method, we perform BaSS operations on one of the augmented input data $x$ to obtain $x^b$ and then perform AugMix to obtain $x^b_{augmix1}$. The model training process constrains the consistency of $x$, $x^b_{augmix1}$, and $x_{augmix2}$.

**Experiment Settings:** We conduct experiments on CIFAR10 and CIFAR100, and utilize CIFAR10-B and CIFAR100-B in training process as mentioned above. We use various architectures including ResNet-50, WideResNet-40-2, DenseNet-29 [22], and ResNeXt [42]. During the testing phase, we

Table 4: **AugMix improves both clean accuracy and corruption robustness combined with BaSS.**

| Dataset | Model | Method | Clean(%) | Robust Acc (%) | | | | |
|---------|-------|--------|----------|------|------|---------|---------|------|
| | | | | *Noise* | *Blur* | *Weather* | *Digital* | *Mean* |
| **CIFAR10** | WRN-40-2 | AugMix | 95.66 | 84.62 | 91.31 | 91.95 | **90.90** | 89.91 |
| | | +BaSS | **95.83** | **88.66** | **91.37** | **93.04** | 90.77 | **90.96** |
| | DenseNet | AugMix | 95.79 | 83.08 | 90.54 | 91.72 | 90.00 | 89.05 |
| | | +BaSS | **95.99** | **88.36** | **91.47** | **93.14** | **90.53** | **90.87** |
| | ResNeXt-29 | AugMix | **96.29** | 82.99 | 91.63 | 92.62 | **91.38** | 89.94 |
| | | +BaSS | 96.08 | **89.37** | **92.10** | **93.43** | 90.94 | **91.44** |
| | ResNet-50 | AugMix | 95.95 | 85.15 | 91.56 | 92.21 | 91.37 | 90.29 |
| | | +BaSS | **96.00** | **89.85** | **91.90** | **93.27** | **91.53** | **91.64** |
| **CIFAR100** | WRN-40-2 | AugMix | 77.53 | 55.43 | **69.08** | 68.24 | **67.13** | 65.41 |
| | | +BaSS | **77.90** | **60.38** | 69.07 | **70.74** | 66.92 | **66.91** |
| | DenseNet | AugMix | 77.46 | 51.83 | 67.74 | 67.49 | 65.19 | 63.53 |
| | | +BaSS | **78.44** | **59.08** | **68.89** | **70.31** | **65.93** | **66.19** |
| | ResNeXt-29 | AugMix | 78.68 | 55.93 | 70.12 | 68.82 | 67.92 | 66.16 |
| | | +BaSS | **80.17** | **60.02** | **71.27** | **72.47** | **68.38** | **68.24** |
| | ResNet-50 | AugMix | 79.01 | 58.90 | 71.26 | 69.95 | 69.74 | 67.90 |
| | | +BaSS | **80.83** | **64.95** | **72.24** | **73.33** | **69.96** | **70.21** |

evaluated the clean accuracy on the test sets of CIFAR10 and CIFAR100, and assessed the corruption robustness on CIFAR10-C and CIFAR100-C [20] datasets.

**Results**. Our method further improves the performance of AugMix as shown in Tab. 4, achieving a better trade-off between generalization and robustness across all model structures. Different classes of corruption usually have their own frequency-domain characteristics [43], thus leads to a bias towards a certain type of corruption in existing defense methods (e.g., adversarial training and AugMix). Notably, our method significantly enhances the model's capabilities in defending against high-frequency patterns corruption (e.g., improving Noise corruption robustness by approximately 4%). This improvement enables the model to achieve a more balanced defense, indicating that our method has indeed enhanced in learning HFC.

We noticed that the BaSS strategy did not help the model's robustness in the "Digital" column of the in Tab. 4. Different types of perturbations usually act on different frequency bands of the image[43], bringing the challenge of diversity of frequency domain information. The Fourier heatmaps [43] show that the type of "Digital" corruption have a common feature: irregular range perturbations exist from low to high frequencies in the image. AugMix resists the above-mentioned irregular frequency band perturbations more efficiently by adopting random data augmentation strategies, producing images with better diversity in frequency domain information. The BaSS method we provided replaced one of the random data augmentation branches in AugMix, thus affecting the original performance of AugMix under the "Digital" type perturbations to some extent.

## 5 Conclusions

In this paper, we have gained a deeper understanding of the relationship between high frequency components (HFC) and model robustness by defining the frequency long-tailed problem. Observations and analyses have revealed that (i) the high sensitivity to HFC stems from the model's under-fitting behavior of HFC, (ii) the cause of the model's under-fitting behavior is attributes to the limited information content of HFC. Based on these insights, we have proposed a balanced spectrum sampling strategy (BaSS) that effectively improves the trade-off between robustness and accuracy when combined with different training paradigms. Our current results are mainly focused on the image classification tasks while we believe it is also rewarding to develop new techniques to other vision tasks. We also expect that this work could inspire a future explanations and defense method from model perspective such as, exploring model-side improvement that can work with BaSS strategy.

## Acknowledgments

This work is supported by the Beijing Natural Science Foundation (No.JQ20023) and the National Natural Science Foundation of China (No. 61832002).

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
