# Supplementary Materia:Revisiting Visual Model Robustness: A Frequency Long-Tailed Distribution View

**Zhiyu Lin**[1], **Yifei Gao**[1], **Yunfan Yang**[1], **Jitao Sang**[1,2*]
[1]Beijing Jiaotong University, China
[2]Peng Cheng Lab, Shenzhen 518066, China
{zyllin, yf-gao, yfyang, jtsang}@bjtu.edu.cn

## A   Related Work

**Understanding Robustness in Frequency Domain.**   Previous research has provided many intriguing insights into model robustness through the lens of frequency spectrum for the Deep Learning (DL) community. Many of these studies attempted to elucidate the connection between model sensitivity and frequency components. A commonly-held hypothesis is that the utilization of high-frequency components leads to a decrease in model robustnessWang et al. [2020]. Recent research has improved model robustness through a variety of methods, guided by this hypothesis. For example, Addepalli et al. [2022] applies regularization terms to high-frequency components to enhance model robustness. Fan et al. [2021] incorporates high-frequency views into contrastive learning, leading to the transfer of pre-trained robust knowledge to downstream tasks. Additionally, Saikia et al. [2021] introduces a high-frequency expert model to ensure robustness to out-of-distribution data. However, there are also several works that challenge the validity of this assumption. Yin et al. [2019] proposes a robustness analysis strategy based on Fourier Heatmaps, which utilizes a model's sensitivity to frequency-bases. They have demonstrated that adversarial attacks are not a strict high-frequency phenomenon. Maiya et al. [2021] believes that model robustness does not have an intrinsic connection with high-frequency components, and their perspective is supported by numerous cross-dataset and cross-model experiments. The lack of a unified explanation within the DL community highlights the need for further investigation. In addition to the perspective on frequency components, Chen et al. [2021] has shown that the CNN model should be consistent with the Human Visual System, with model robustness increased by enhancing the utilization of the phase spectrum.

**Long-tailed Recognition.**   Visual long-tailed recognition tasks typically involve addressing the inter-class long-tailed effect, which is often achieved through resampling He and Garcia [2009], Shen et al. [2016] and reweighting Cui et al. [2019] methods. These methods aim to re-balance the training set and ensure that each class is given appropriate consideration. However, in real-world scenarios, the long-tailed effect may be more reflected in intra-class attributes Tang et al. [2020], where rare attributes or samples within each class are often difficult to classify accurately. Therefore, improving the generalization of models on intra-class attributes is also an important issue. Additionally, some works have also investigated model robustness under long-tailed distributions. Wu et al. [2021] first revealed the recognition performance of imbalanced data pairs and the negative effects of adversarial robustness, and then combined them with the adversarial training framework to conduct a systematic study on the existing long-tailed recognition methods.

---

*Corresponding Author

37th Conference on Neural Information Processing Systems (NeurIPS 2023).

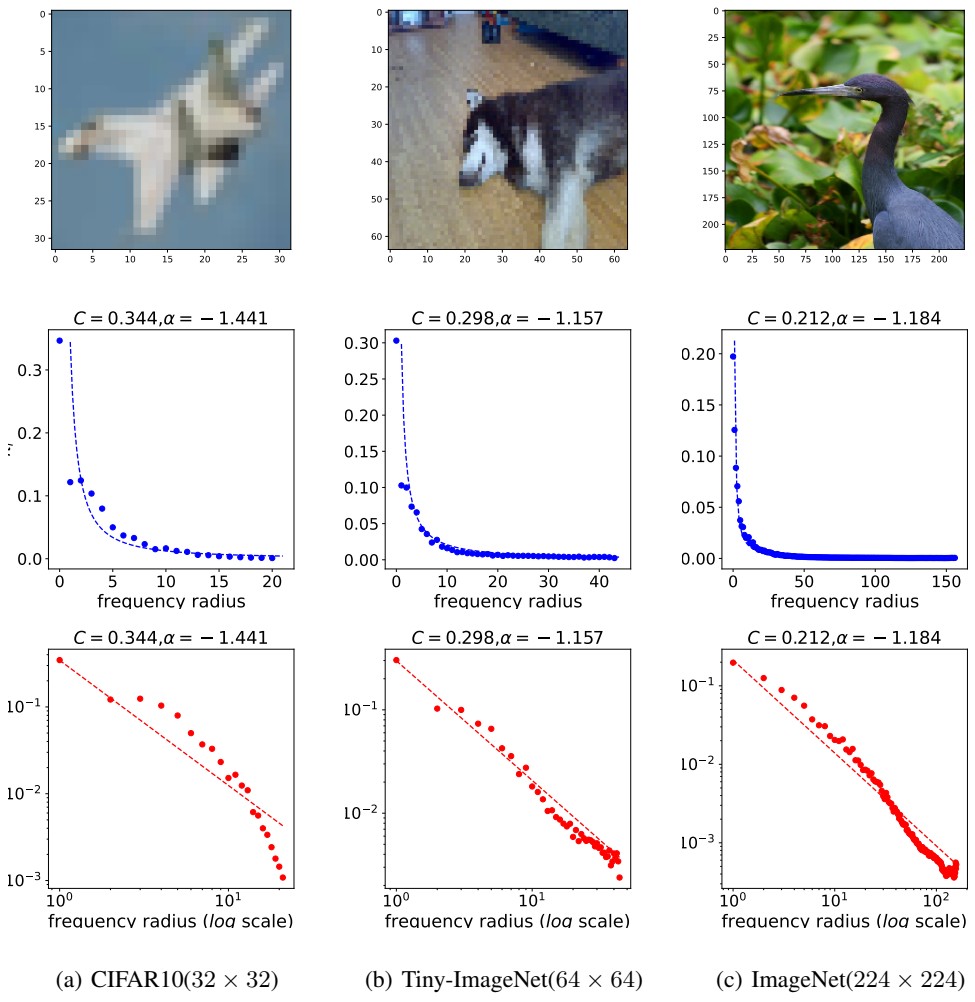

Figure 1: The power law distribution within frequency domain of images.

## B    Details on definition of frequency long tailed problem

### B.1    Power law distribution of natural images

To show the power law distribution of natural images, we select CIFAR-10 Krizhevsky et al. [2009], Tiny-ImageNet Le and Yang [2015] and ImageNet Deng et al. [2009] to conduct experiments. We first perform 2D discrete Fourier transform $\mathcal{F}$ on images $X$ to get spectrum $\tilde{X}$, and calculate power spectral density $AI(r_k)$ and its proportion $\pi_k$ for $k^{th}$ frequency band, the power distribution result is shown in the middle sub-figure in Fig. 1. To show the distribution follows the power law distribution $p(x) = Cx^{-\alpha}$, we utilize the data point of distribution curve to fit the parameter $C$ and $\alpha$. We further sample the data point to plot the dash line, and display the curve with log-log scale coordinate in the last row in Fig. 1. On the data sets of different resolutions, we observed that in both normal and log-log scale coordinate systems, the image frequency power conform to the power law distribution, within the allowable range of error.

### B.2    Spectrum Division

We show an example of division on ImageNet, as shown in Fig.2, in which the high- and low-frequency components of the image obtained according to the division radius are also in line with our

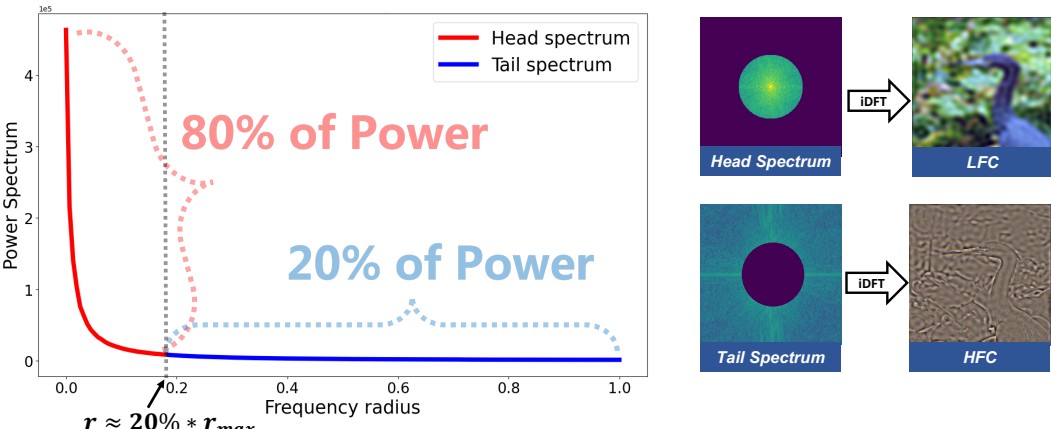

Figure 2: An example of spectrum division based on Pareto principle.

qualitative understanding of frequency components from the visualization results, compared with the manually setting the radius , we provide a quantifiable and stable basis.

## C Visualization of Loss Landscape on Frequency Components

We first split the images in test set into low- and high-frequency components, denoted as $X_l$ and $X_h$, respectively.

$$
\begin{cases}
X_l^{(r)} = \mathcal{F}^{-1}(\mathcal{F}(X) \odot \mathcal{M}_l^{(r)}) \\
X_h^{(r)} = \mathcal{F}^{-1}(\mathcal{F}(X) \odot \mathcal{M}_h^{(r)})
\end{cases}
\tag{1}
$$

where $\mathcal{F}^{-1}(\cdot)$ denote the 2D-inverse discrete Fourier transform, $\odot$ denotes the Hadamard product and $\mathcal{M}^{(r)}$ denotes the matrix of characteristic function with frquency radius $r$:

$$
\mathcal{M}_l^{(r)}(i,j) =
\begin{cases}
1, & if \ \ d((i,j),(c_i,c_j)) \leq r \\
0, & otherwise
\end{cases}
\tag{2}
$$

where $d(\cdot,\cdot)$ denotes the Euclidean distance from spectrum coordinate $(i,j)$ to the center of spectrum $(c_i,c_j)$. Thus, we have $\mathcal{M}_h^{(r)} = 1 - \mathcal{M}_l^{(r)}$.

We conduct experiments on naturally trained models. The model parameters are denoted as the the center point $\theta$, we choose two direction vector, $\delta$ and $\eta$ to plot function of $f(\alpha,\beta) = \mathcal{L}(X,y,\alpha\theta+\beta\eta)$. Furthermore, the frequency-specific plot function is denoted as $f_l(\alpha,\beta) = \mathcal{L}(X_l,y,\alpha\theta + \beta\eta)$ and $f_h(\alpha,\beta) = \mathcal{L}(X_h,y,\alpha\theta + \beta\eta)$, respectively.

## D Analyzing Model Sensitivity to Frequency Components

**Understanding Estimation Methods.** In the 2-D discrete Fourier transform, the Fourier basis vector $\omega_{u,v}$ can be understood as a two-dimensional plane wave controlled by the directions $u$ and $v$, as shown in Fig. 3(a). As the values of $u$ and $v$ change, the oscillation direction of the basis vector will differ. Therefore, we use vector $(u,v)$ to describe the characteristics of the basis vector $\omega_{u,v}$. Our method projects the gradient into the space composed of the aforementioned orthogonal Fourier basis vectors, and uses the projection coordinates $c_{u,v}$ to represent the magnitude of the gradient projection. Since the Fourier transform of the input image $X$ and the gradient share the same Fourier space, we can measure the model's sensitivity to different frequency domain directions of the input image by comparing the gradient magnitudes on different basis vectors. Fig. 3(b) shows the results of applying the Fourier transform to the image gradients in ImageNet dataset. It is clear to see that the gradients present a smooth and regular distribution in different frequency domain directions, with the gradient magnitude mainly concentrated in the high-frequency direction of the input image (i.e., on the four corners of the spectrum).

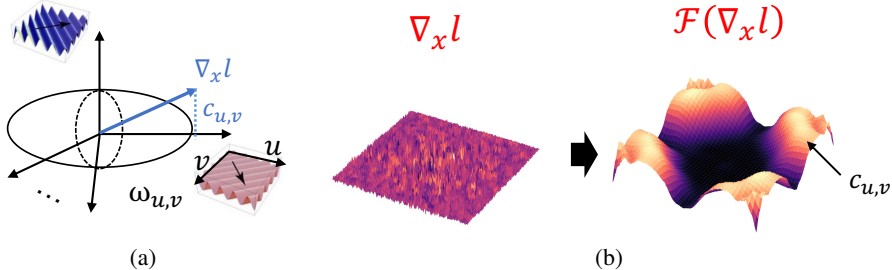

(a)                                                         (b)

Figure 3: Schematic diagram of gradient projection into Fourier space

**Experiments Settings.** We conduct experiments on test set of CIFAR10, Tiny-ImageNet, ImageNet-1k datasets. We trained ResNet-18 He et al. [2016] for CIFAR10 and Tiny-ImageNet, and utilized the pre-trained ResNet-50 model for ImageNet. For the local input gradient $g$, we average the value on each channel and perform 2-D discrete Fourier transform. By sorting the Fourier basis vector with gradient magnitude $\|c_{u,v}\|_2$, we obtain the set of feature points $\{r_i\}_{i=1}^{N}$ in $k^{th}$ rank index of all images, where $N$ is the number of images. We estimate the kernel density on feature $r$ as follows:

$$S_k(r) = \frac{1}{N} \sum_{i=1}^{N} K(r - r_i) = \frac{1}{Nh} \sum_{i=1}^{N} K(\frac{r - r_i}{h}) \tag{3}$$

where $K(\cdot)$ denotes the Gaussian kernel function, $h$ is the bandwidth calculated by Silverman's rule. To show the results, we select the value of $k$ in the set of $\{1, 0.2 \times r_{max}, 0.4 \times r_{max}, 0.6 \times r_{max}, 0.8 \times r_{max}, r_{max}\}$, where $r_{max}$ is the maximum of rank index.

**Results on ViT Backbone.** To further explore the phenomenon on vision transformer backbone, we selected the pre-trained model Vit-B/16 Dosovitskiy et al. [2020] and Vit-B/32 trained on ImageNet. Considering that in the ViT model, images are fed into the model in the form of patches, we calculate gradients on each patch and then perform frequency-domain spatial projections, and represent the frequency-domain sensitivity of this image with the mean gradient magnitude on all patches. As shown in Fig. 4, we observed that the sensitivity of the model was concentrated in the tail high-frequency band of the image patch, which is consistent with the experimental results on the CNN structure, which further supports our conclusions. At the same time, it shows that the frequency long-tailed problem mainly stems from the images spectrum characteristics, so it leads to consistent phenomena in different model structures, which also inspires us to address the frequency long-tailed problem from the data side.

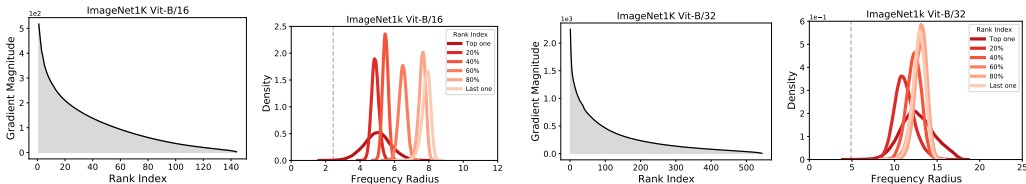

(a) Vit-B/16 model sensitivity to ImageNet-1k dataset (b) Vit-B/32 model sensitivity to ImageNet-1k dataset

Figure 4: Model sensitivity results on ViT backbone and ImageNet-1k dataset.

# E   Proof of Theorem

**Theorem 1** *Given any natural image $X$, $k$ represents the order of frequency band, then the spectral entropy $H_k$ is given by: $H_k = (\alpha A_\alpha \log k + H(A(\alpha))) \cdot k^{-\alpha}$, where $A(\alpha) = (1 - \alpha) \cdot (\xi_h^{1-\alpha} - \xi_l^{1-\alpha})^{-1}$. Then we have, $\lim_{k \to 0} H_k = \infty$ and $\lim_{k \to \infty} H_k = 0$.*

**Proof E.1** *With assumption that the spectral density of a natural image follows:* $p(\xi = k) = \frac{1-\alpha}{\xi_h^{1-\alpha} - \xi_l^{1-\alpha}} \cdot k^{-\alpha}$, *then we have*

$$
\begin{aligned}
H_k &= -p(\xi = k) \log p(\xi = k) \\
&= -A(\alpha) \cdot k^{-\alpha} \cdot \log(A(\alpha) \cdot k^{-\alpha}) \\
&= -A(\alpha) \cdot k^{-\alpha} \cdot (\log A_\alpha - \alpha \log k) \\
&= \frac{\alpha A(\alpha) \cdot \log k}{k^\alpha} + \frac{-A(\alpha) \log A(\alpha)}{k^\alpha} \\
&= \frac{\alpha A(\alpha) \cdot \log k + H(A(\alpha))}{k^\alpha}
\end{aligned}
$$

*Therefore, we have* $\lim_{k \to \infty} H_k = \alpha A(\alpha) \cdot \lim_{k \to \infty} \frac{\log k}{k^\alpha} = \lim_{k \to \infty} (\alpha k^\alpha)^{-1} = 0$, *and* $\lim_{k \to 0} H_k = \infty$, *where* $A(\alpha)$ *denotes* $\frac{1-\alpha}{\xi_h^{1-\alpha} - \xi_l^{1-\alpha}}$. *Obviously, the spectral entropy decreases at an* $\alpha$ *order.*

**Theorem 2** *Given the prior data* $\xi$, *and its corresponding probability distribution* $p(\xi)$, *the principle of maximum entropy consider the following optimization:* $\hat{p} = \{arg \max_{p \in \mathbb{P}} \ H(p) \mid p(\xi) \geq 0, \int_\Omega p(\xi) d\xi = 1, \int_\Omega p(\xi) r_i(\xi) d\xi = \alpha_i, i = 1, \cdots m\}$. *Without considering the moment constraints, the uniform distribution* $p(\xi) = \frac{1}{\xi_h - \xi_l} 1_{[\xi_l, \xi_h]}(\xi)$ *satisfies the principle of maximum entropy.*

**Proof E.2** *The principal of maximum entropy considers the following optimization,* $\hat{p} := arg \max_{p \in \mathbb{P}} H(p)$, *where* $p$ *satisfies the following constraints:* $p > 0$, $\int_\Omega p(\xi) d\xi = 1$ *and* $\int_\Omega p(\xi) r_i(\xi) d\xi = \alpha_i, i = 1, \cdots m$, *where* $\Omega$ *is the range in* $[\xi_l, \xi_h]$, *and the latter are also commonly referred to as moment constraints. Without considering the moment constraints, the Lagrange function is given by*

$$
\mathcal{J}(p) = H(p) + \lambda\left(\int_\Omega p(\xi) d\xi - 1\right) = -\int_\Omega p(\xi) \log p(x) dx + \lambda\left(\int_\Omega p(\xi) d\xi - 1\right)
$$

*where* $\lambda$ *is Lagrange multiplier. Then the functional differential is given by* $\frac{\partial \mathcal{J}(p)}{\partial p} = -\log p - 1 + \lambda$. *Let the partial derivatives be zero, then we have* $p = e^{\lambda - 1}$. *Considering the constraint* $\int_\Omega p(\xi) d\xi = 1$, *then we have* $p(\xi) = \frac{1}{\xi_h - \xi_l} 1_{[\xi_l, \xi_h]}(\xi)$, *which satisfies the principle of maximum entropy.*

## F  Details on Methodology and Experiments

### F.1  Balanced Spectrum Sampling(BaSS) Methodology

---
**Algorithm 1:** Balanced Spectrum Sampling(BaSS)

---
**Input:** The natural image $X$; parameter $\tau$.
**Output:** The spectrum balanced image $X_B$.

1 **foreach** *image channel c* **do**
2     $\tilde{X}_c \leftarrow \mathcal{F}(X_c)$;
3     **foreach** *frequency band i* **do**
4        $\pi_i \leftarrow \frac{AI(r_i)}{\sum_{j=1}^K AI(r_j)}$;
5        $\hat{\pi}_i \leftarrow \frac{\log_\tau AI(r_i)}{\sum_{j=1}^K \log_\tau AI(r_j)}$;
6        $Weight \leftarrow \frac{\pi_i}{\hat{\pi}_i}$;
7        $\tilde{X}_c^i \leftarrow \tilde{X}_c^i \cdot Weight$;
8     **end**
9     $X_c \leftarrow \mathcal{F}^{-1}(\tilde{X}_c)$;
10 **end**
11 Output $X_B$ by updating each channel $X_c$.

---

The algorithm for proposed BaSS is shown in Algo. 11. It is worth noting that in the experiment, we uniformly set the value of $\tau$ as $e$ on different datasets. In future work, we believe that the value of $\tau$ should be adaptive to images of different resolutions, because considering that in different degrees of long-tail distribution, the unbalanced relationship between the head and the tail will be different, so it needs to be based on the characteristics of spectrum power density to choose the value of $\tau$.

We illustrate the results of images and spectral power density when performing BaSS in Fig. 6. The spectral power density curve was calculated across all images in the test set. Comparing the images shows that images after spectrum re-sampling have a richer high-frequency component, such as contours and texture information are enhanced, while some low-frequency information is also retained in the images. Unlike methods that extract contours or textures and retain only high-frequency information, completely filtering out low-frequency information, BaSS achieves a balance between low-frequency and high-frequency components. This can be corroborated by the spectral power density curve, where the power distribution across all frequency bands is more balanced after sampling, as seen in the under-sampling of the low frequencies and over-sampling of the high-frequencies distribution. Compared to the original power-law distribution, the power curve after sampling is closer to a linear distribution. This validates the two starting points of our method: (i) balance the power density of different frequency bands, and (ii) maintain the relative relationship between low frequencies and high frequencies, avoiding having equal power distribution on ultra-high frequency noise and other frequency bands.

## F.2 Details on Experiments.

**Datasets.** To evaluate our proposed method BaSS, we choose CIFAR10, CIFAR100, Restricted-ImageNet datasets to conduct experiments. CIFAR10 and CIFAR100 are benchmark datasets for evaluating model robustness, where CIFAR100 contains a larger number of categories, so it is more challenging for model generalization and robustness evaluation. At the same time, in order to verify that our method is also effective in higher image resolutions, we set two resolution images based on the Restricted-ImageNet dataset, $64 \times 64$ and $128 \times 128$, respectively, without changing the number of images in the dataset.

### F.2.1 Details on Woking Mechanism Exploration of BaSS

**Training Settings.** We trained ResNet-18 on CIFAR10, CIFAR10-B (i.e., consists of images performing BaSS on CIFAR10) and mixed datasets (i.e., consists of images both from CIFAR10 and CIFAR10-B). For mixed datasets training, we perform BaSS on images in each training epoch with a probability of $\gamma$ and sampling follows uniform distribution. For all experiments, we set 200 training epochs and use SGD optimizer with cosine learning rate schedule with a momntum of 0.9, the initial learning rate of 0.1 and weight decay of 5e-4.

**It is worth considering how to combine CIFAR10 and CIFAR10-B.** From the results in Tab. 8 and Tab. 10, we observe a significant improvement in both model corruption robustness and adversarial robustness when incorporating CIFAR10-B. At the same time, the model maintains a high accuracy on clean samples. This indicates the effectiveness of using the CIFAR10-B dataset for data augmentation. Additionally, we notice that as the proportion of CIFAR10-B increases, there is a decreasing trend in accuracy on CIFAR10, especially when $\gamma = 1.0$. The model exhibits lower generalization on CIFAR10, and we attribute this phenomenon to the out-of-distribution (OOD) relationship between the sampled images and the original images in terms of data distribution. This OOD relationship is caused by the re-sampling operation on the image spectrum. Therefore, in future work, we will consider how to better integrate the two datasets.

**Information from various frequency bands of an image is required to enhance the corruption robustness.** From the results in Tab. 9, we observe that for the CIFAR10-B dataset, using CIFAR10 as data augmentation also leads to noticeable improvement in robustness. However, the effects of the two datasets on different corruption categories are inconsistent. For example, CIFAR10-B shows significant improvement in handling high-frequency $noise$ perturbations, while CIFAR10 exhibits noticeable improvement in dealing with $blur$ type corruption. Typically, different corruption types exhibit distinct frequency spectrum patterns in the frequency domain. This insight inspires us to enhance the corruption robustness by utilizing image datasets with richer frequency domain

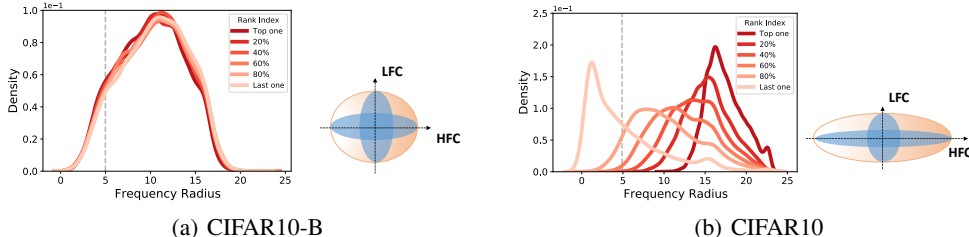

(a) CIFAR10-B           (b) CIFAR10

Figure 5: Comparison of sensitivity to frequencies between model trained on CIFAR10-B and CIFAR10, respectively.

information. Otherwise, the model may suffer from bias issues in robustness under certain perturbation categories.

**What kind of data is beneficial for improving model's adversarial robustness?** Comparing the results of Tab. 10 and Tab. 11, we find that the adversarial robustness of the models trained and tested on CIFAR10-B is significantly better than that of CIFAR10. This inspires us to explore the reasons behind it. We projected the data gradients of CIFAR10 and CIFAR10-B into the frequency domain space and visualized them as shown in the Fig. 5. Interestingly, we discovered that the gradients of the model on CIFAR10-B are distributed more evenly across different frequency bands, rather than being primarily concentrated on high frequencies like CIFAR10. This inspires us to consider a strategy for reducing the model's adversarial risk by constraining the model to evenly distribute gradients across different dimensions of the data during the learning process, so that each dimension of the data has a balanced feature representation. Natural images, due to their inherent imbalanced distribution, lead to bias phenomena during the learning process, causing certain directions to have excessively large gradients while the remaining directions collapse. On the other hand, models trained on CIFAR10-B exhibit a fuller distribution in the data gradient space, making it less susceptible to adversarial attacks along a particular direction.

### F.2.2 Details on BaSS Working in Conjunction with Adversarial Training

**Experiments Settings and Results.** The detailed hyperparameters of the adversarial training process are shown in the Tab. 6 and Tab. 7. We supplement the results on the CIFAR10 and CIFAR100 datasets in Tab. 1,2,3 over 5 random runs. We show the results of PGD-AT and TRADES combined with BaSS on the Restricted-ImageNet dataset in Tab. 4. We compare the training overhead of our method with standard adversarial training in Tab. 5.

### F.2.3 Details on BaSS Working in Conjunction with AugMix

We trained ResNet-50, WideResNet-40-2, DenseNet-29 Huang et al. [2017], and ResNeXt Xie et al. [2017] on CIFAR10 and CIFAR100, and use CIFAR10-B and CIFAR100-B dataset as data augmentation. We evaluate corruption robustness on CIFAR10-C and CIFAR100-C. For all experiments, we set 200 training epochs and use SGD optimizer with cosine learning rate schedule with a momntum of 0.9, the initial learning rate of 0.1 and weight decay of 5e-4.

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

Table 1: **Clean and Robust Accuracy (%) of PGD-AT and PGD-AT with BaSS on CIFAR-10,CIFAR-100 using ResNet-18 over 5 random runs.**

| Dataset | Method | Clean Acc | Robust Acc | | |
| --- | --- | --- | --- | --- | --- |
| | | | *PGD-AT* | *CW* | *AA* |
| CIFAR10 | PGD-AT | 83.50±0.18 | 52.72±0.21 | 51.15±0.16 | 48.90±0.12 |
| | PGD-AT+BaSS | **89.22±0.13** | **59.61±0.27** | **58.62±0.18** | **55.90±0.08** |
| CIFAR100 | PGD-AT | 56.33±0.20 | 29.29±0.17 | 27.66±0.18 | 24.88±0.13 |
| | PGD-AT+BaSS | **63.91±0.21** | **31.61±0.23** | **30.70±0.11** | **27.95±0.18** |

Table 2: **Clean and Robust Accuracy (%) of TRADES and TRADES with BaSS on CIFAR-10,CIFAR-100 using ResNet-18 over 5 random runs.**

| Dataset | Method | Clean Acc | Robust Acc | | |
| --- | --- | --- | --- | --- | --- |
| | | | *PGD* | *CW* | *AA* |
| CIFAR10 | TRADE | 82.15±0.22 | 52.50±0.31 | 50.15±0.27 | 49.05±0.16 |
| | TRADE+BaSS | **87.20±0.16** | **60.01±0.28** | **57.52±0.24** | **56.27±0.07** |
| CIFAR100 | TRADE | 58.30±0.26 | 29.90±0.22 | 26.68±0.16 | 25.52±0.19 |
| | TRADE+BaSS | **64.17±0.17** | **32.94±0.27** | **29.01±0.19** | **27.77±0.21** |

Table 3: **Clean and Robust Accuracy (%) of MART and MART with BaSS on CIFAR-10,CIFAR-100 using ResNet-18 over 5 random runs.**

| Dataset | Method | Clean Acc | Robust Acc | | |
| --- | --- | --- | --- | --- | --- |
| | | | *PGD* | *CW* | *AA* |
| CIFAR10 | MART | 81.38±0.28 | 54.74±0.19 | 51.35±0.21 | 49.48±0.19 |
| | MART+BaSS | **88.75±0.11** | **60.43±0.21** | **58.75±0.23** | **56.93±0.25** |
| CIFAR100 | MART | 57.8±0.19 | 30.73±0.33 | 29.97±0.26 | 26.26±0.23 |
| | MART+BaSS | **63.94±0.16** | **32.95±0.27** | **31.92±0.31** | **28.01±0.25** |

Table 4: **Clean and Robust Accuracy(%) on Restricted-ImageNet dataset.**

| Method | **Restricted-ImageNet**$(64 \times 64)$ | | | **Restricted-ImageNet**$(128 \times 128)$ | | |
| --- | --- | --- | --- | --- | --- | --- |
| | *Clean* | $PGD^{10}$ | *AA* | *Clean* | $PGD^{10}$ | *AA* |
| PGD-AT | 65.13 | 49.44 | 33.37 | 72.81 | 59.83 | 40.35 |
| +BaSS | 66.76 | 57.63 | 41.77 | 73.12 | 63.63 | 45.90 |
| $\Delta$ | **+1.63** | **+8.19** | **+8.40** | **+0.31** | **+3.80** | **+5.55** |

Table 5: **Training time per epoch (Seconds) of PGD-AT and our methods on a single Nvidia A6000 GPU.**

| Dataset | Model | PGD-AT | +BaSS |
| --- | --- | --- | --- |
| CIFAR10$(32 \times 32)$ | WRN-34-10 | 1474 | 1494 |
| CIFAR100$(32 \times 32)$ | | 1500 | 1520 |
| Restricted-ImageNet$(64 \times 64)$ | ResNet-50 | 1352 | 1410 |
| Restricted-ImageNet$(128 \times 128)$ | | 2334 | 2352 |

Table 6: **Hyperparameters for adversarial training .**

| Dataset | Epochs | LR | LR Drop(Epoch) | Weight Decay | Batch Size |
|---|---|---|---|---|---|
| CIFAR10($32 \times 32$)
CIFAR100($32 \times 32$) | 110 | 0.1 | 0.1(100,105) | 5e-4 | 128 |
| Restricted-ImageNet($64 \times 64$)
Restricted-ImageNet($128 \times 128$) | 100 | 0.1 | 0.1(30,60,90) | 1e-4 | 256 |

Table 7: **PGD parameters used to construct adversarial examples in adversarial training.**

| Dataset | $\epsilon$ | Step Size | Iterations |
|---|---|---|---|
| CIFAR10($32 \times 32$)
CIFAR100($32 \times 32$) | $\frac{8}{255}$ | $\frac{2}{255}$ | 10 |
| Restricted-ImageNet($64 \times 64$)
Restricted-ImageNet($128 \times 128$) | $\frac{4}{255}$ | $\frac{1}{255}$ | 10 |

Table 8: **Clean Accuracy(%) on CIFAR10 and Robust Accuracy(%) on CIFAR10-C.**

| Training Datasets | | CIFAR10 | Mixed Datasets($\gamma$) | | | CIFAR10-B |
|---|---|---|---|---|---|---|
| | | $\gamma = 0.0$ | $\gamma = 0.25$ | $\gamma = 0.5$ | $\gamma = 0.75$ | $\gamma = 1.0$ |
| *Noise* | gaussian | 44.71 | 68.52 | 68.04 | 66.31 | 43.86 |
| | shot | 57.77 | 71.22 | 67.95 | 73.85 | 46.45 |
| | impulse | 56.11 | 75.69 | 75.44 | 66.15 | 44.52 |
| | speckle | 61.79 | 77.61 | 77.17 | 75.22 | 46.54 |
| | Mean | 55.10 | **73.26** | 72.15 | 70.38 | 45.34 |
| *Blur* | defocus | 81.54 | 88.20 | 89.33 | 89.78 | 29.67 |
| | glass | 52.02 | 71.48 | 70.84 | 69.77 | 38.04 |
| | motion | 77.92 | 81.38 | 81.84 | 81.74 | 24.99 |
| | zoom | 77.37 | 85.54 | 86.51 | 87.45 | 24.54 |
| | gaussian | 71.11 | 83.41 | 85.05 | 86.80 | 26.87 |
| | Mean | 71.99 | 82.00 | 82.71 | **83.11** | 28.22 |
| *Weather* | snow | 83.11 | 86.70 | 86.42 | 86.08 | 42.32 |
| | frost | 77.55 | 87.58 | 88.07 | 87.76 | 48.65 |
| | fog | 88.51 | 90.66 | 91.09 | 90.91 | 32.22 |
| | brightness | 94.19 | 94.20 | 94.22 | 94.02 | 51.72 |
| | Mean | 85.84 | 89.79 | **89.95** | 89.69 | 43.73 |
| *Digital* | contrast | 77.72 | 79.53 | 79.25 | 77.56 | 33.76 |
| | elastic | 84.78 | 87.31 | 86.95 | 86.62 | 29.06 |
| | jpeg | 79.54 | 81.11 | 80.95 | 80.78 | 33.27 |
| | pixelate | 76.66 | 82.22 | 79.75 | 82.82 | 44.51 |
| | saturate | 92.47 | 92.56 | 92.24 | 91.99 | 47.01 |
| | spatter | 85.99 | 88.60 | 88.08 | 87.45 | 46.74 |
| | Mean | 82.86 | **85.22** | 84.54 | 84.54 | 39.06 |
| *Robust Acc* | | 74.78 | **82.82** | 82.59 | 82.27 | 38.67 |
| *Clean Acc* | | **95.63** | 95.47 | 95.2 | 94.88 | 50.12 |

Table 9: **Clean Accuracy(%) on CIFAR10-B and Robust Accuracy(%) on CIFAR10-B-C.**

| Training Datasets | | CIFAR10 | Mixed Datasets($\gamma$) | | | CIFAR10-B |
|---|---|---|---|---|---|---|
| | | $\gamma = 0.0$ | $\gamma = 0.25$ | $\gamma = 0.5$ | $\gamma = 0.75$ | $\gamma = 1.0$ |
| *Noise* | gaussian | 31.87 | 83.82 | 83.27 | 81.39 | 83.46 |
| | shot | 37.4 | 86.58 | 86.24 | 84.91 | 86.26 |
| | impulse | 33.53 | 79.49 | 78.30 | 77.09 | 77.93 |
| | speckle | 37.93 | 86.65 | 86.30 | 84.96 | 86.43 |
| | Mean | 35.18 | **84.14** | 83.53 | 82.09 | 83.52 |
| *Blur* | defocus | 68.59 | 88.38 | 89.04 | 89.51 | 75.17 |
| | glass | 24.46 | 61.53 | 61.34 | 59.63 | 57.73 |
| | motion | 45.44 | 76.81 | 77.55 | 77.32 | 67.99 |
| | zoom | 71.94 | 84.11 | 85.08 | 85.69 | 69.24 |
| | gaussian | 79.14 | 89.72 | 90.34 | 90.66 | 62.93 |
| | Mean | 57.91 | 80.11 | **80.67** | 80.56 | 66.61 |
| *Weather* | snow | 55.19 | 90.64 | 91.25 | 91.03 | 89.99 |
| | frost | 55.11 | 91.65 | 92.15 | 91.88 | 91.03 |
| | fog | 59.24 | 89.54 | 90.73 | 91.14 | 83.98 |
| | brightness | 62.05 | 92.87 | 93.55 | 93.18 | 92.92 |
| | Mean | 57.90 | 91.18 | **91.92** | 91.81 | 89.48 |
| *Digital* | contrast | 51.64 | 82.23 | 83.35 | 83.04 | 75.44 |
| | elastic | 71.89 | 86.62 | 86.86 | 87.00 | 76.03 |
| | jpeg | 45.14 | 86.13 | 86.85 | 86.30 | 86.83 |
| | pixelate | 58.38 | 86.97 | 86.36 | 85.80 | 87.61 |
| | saturate | 57.76 | 91.09 | 91.18 | 91.01 | 90.25 |
| | spatter | 55.39 | 91.27 | 91.73 | 91.45 | 91.29 |
| | Mean | 56.70 | 87.39 | **87.72** | 87.43 | 84.58 |
| *Robust Acc* | | 52.74 | 85.58 | **85.86** | 85.42 | 80.66 |
| *Clean Acc* | | 63.15 | 94.02 | **94.58** | 94.15 | 94.13 |

Table 10: **Clean Accuracy(%) on CIFAR10 and Robust Accuracy(%) on PGD-20 attack.**

| Training Datasets | | CIFAR10 | Mixed Datasets($\gamma$) | | | CIFAR10-B |
|---|---|---|---|---|---|---|
| | | $\gamma = 0.0$ | $\gamma = 0.25$ | $\gamma = 0.5$ | $\gamma = 0.75$ | $\gamma = 1.0$ |
| *step size=$\epsilon/4$* | $\epsilon = 1/255$ | 42.64 | 50.21 | 52.18 | **56.17** | 25.31 |
| | $\epsilon = 2/255$ | 9.48 | 15.37 | 16.82 | **19.91** | 10.02 |
| | $\epsilon = 4/255$ | 0.53 | 1.99 | 2.09 | **2.48** | 1.45 |
| | $\epsilon = 8/255$ | 0.02 | **0.14** | 0.11 | 0.12 | 0.02 |
| *Robust Acc* | | 13.17 | 16.93 | 17.80 | **19.67** | 9.20 |
| *Clean Acc* | | **95.63** | 95.47 | 95.2 | 94.88 | 50.12 |

Table 11: **Clean Accuracy(%) on CIFAR10-B and Robust Accuracy(%) on PGD-20 attack.**

| Training Datasets | | CIFAR10 | Mixed Datasets($\gamma$) | | | CIFAR10-B |
|---|---|---|---|---|---|---|
| | | $\gamma = 0.0$ | $\gamma = 0.25$ | $\gamma = 0.5$ | $\gamma = 0.75$ | $\gamma = 1.0$ |
| *step size=$\epsilon/4$* | $\epsilon = 1/255$ | 22.62 | 70.65 | 72.67 | 73.00 | **75.78** |
| | $\epsilon = 2/255$ | 5.61 | 39.01 | 40.86 | 40.63 | **47.39** |
| | $\epsilon = 4/255$ | 0.32 | 8.45 | 8.90 | 9.05 | **12.02** |
| | $\epsilon = 8/255$ | 0.02 | **0.55** | 0.49 | 0.53 | 0.49 |
| *Robust Acc* | | 7.14 | 29.66 | 30.73 | 30.80 | **33.92** |
| *Clean Acc* | | 63.15 | 94.02 | **94.58** | 94.15 | 94.13 |

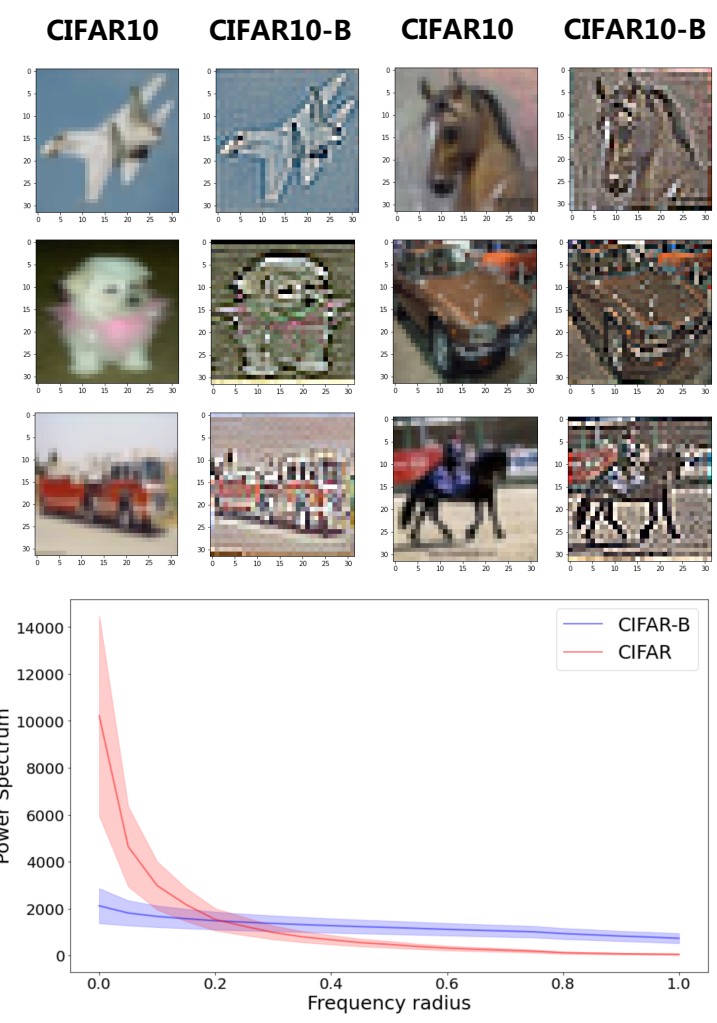

Figure 6: Examples of CIFAR10-B and comparison of power density between CIFAR10 and CIFAR10-B.