# OpenReview forum: "Revisiting Visual Model Robustness: A Frequency Long-Tailed Distribution View"
_NeurIPS.cc/2023/Conference — NeurIPS 2023 poster_

### Official Review · Reviewer_zwAk · 2023-06-29

**Soundness:** 3 good
**Presentation:** 3 good
**Contribution:** 3 good
**Rating:** 7
**Confidence:** 3

**Summary:**

In this paper, the authors investigate the high-frequency components (HFC) in images from a long-tailed perspective. They revisit the relationship between HFC and model robustness and demonstrate that the cause of the model’s under-fitting behavior is attributed to the limited information content in HFC. A Balance Spectrum Sampling (BaSS) strategy, which effectively counteracts the long-tailed effect and enhances the model’s learning on HFC. Extensive experimental results demonstrate the effectiveness.

**Strengths:**

1. Investigating the high-frequency components (HFC) in images from a long-tailed perspective for the model robustness task sounds good and very interesting.
2. The authors give detailed definitions and analysis of the frequency long-tailed scenario.
3. The proposed solution BaSS is simple and efficient.
4. Experiments are conducted on a various benchmarks.


**Weaknesses:**

1. To my end, the re-sampling/re-weighting method in the long-tailed situation generally improves the recognition ability of the tails at the expense of a portion of heads, does this phenomenon will weaken the recognition ability of LFC?
2. Some typos, like:
    line 16, counteract -> counteracts
    line 17, enhance -> enhances


**Questions:**

1. Is this paper the first to analyze the problem of model robustness from the perspective of frequency long-tails?


**Limitations:**

The authors do not address the limitations in this paper.

---

> ### Author Rebuttal · Authors · 2023-08-09
>
> We sincerely appreciate your positive comments for our work. We set out below our responses to each of the questions.
>
> - **Q1: To my end, the re-sampling/re-weighting method in the long-tailed situation generally improves the recognition ability of the tails at the expense of a portion of heads, does this phenomenon will weaken the recognition ability of LFC?**
>
>   - **R1**: Thank you for your question! Firstly, we would like to introduce a phenomenon that we have observed, which is BaSS does indeed result in a certain decline in low-frequency spectral density, as demonstrated in Figure 7 in the Appendix. This observation aligns with the trend of reducing the sampling rate or weight of the head classes. Consequently, it is valuable and intriguing to discuss whether the model's performance on LFC has been weakened or not. We present the following experimental results and would appreciate discussing with the reviewers. We have selected models based on standard training, AugMix, adversarial training, one with BaSS and the other without BaSS, and report their recognition performance on both LFC and HFC.
>
>     |           Method                    | Clean acc on LFC  |  Clean acc on HFC  |
>     | :---------------------------: | :---: | :---: |
>     |      standart training.       | 81.59% | 18.19% |
>     |  **standart training+BaSS**   | 76.77% | 28.84% |
>     |            AugMix             | 85.40% | 23.64% |
>     |        **AugMix+BaSS**        | 82.77% | 34.10%  |
>     |     Adversarial training      | 85.57% | 9.88%  |
>     | **Adversarial training+BaSS** | 84.23% | 17.90% |
>
>     It can be observed that the performance on LFC has shown a decrease trend. This decline can be attributed to BaSS's focus on instance-wise balance, which, in turn, reduces the amount of information available in the low-frequency range. This observation aligns with the analysis presented in section 4. Conversely, it is worth noting that the model has improved the recognition ability on high frequencies. By combining the results from Tables 1 to 4, we can draw the conclusion that performing BaSS benefits the model's recognition on high frequencies and enhances the robust and generalization performance for full image recognition. Nevertheless, it is worth noting that this approach also exhibits some negative impact on low frequencies. This observation serves as inspiration for us to explore more effective data sampling strategies or optimization algorithms on the model side in future endeavors. We have included a detailed discussion of these findings in Appendix F.2.
>
>
>
> - **Q2: Is this paper the first to analyze the problem of model robustness from the perspective of frequency long-tails?**
>
>   - **R2**: Thank you for your question. We would like to confirm that our work represents a pioneering effort in analyzing the robustness and generalizability of vision models from the perspective of frequency long-tails. While some related works have focused on the class- or attribute-wise long-tail problem in image recognition tasks (refer to Line 29-31 in the Appendix), the core contribution of our work lies in introducing a novel instance-wise (frequency) long-tailed problem definition, which also enriches the scene of visual long-tailed tasks. We thoroughly investigate and analyze the impact of the long-tailed spectrum on vision models, and propose an effective solution based on a balanced spectrum sampling strategy to address the challenges posed by the frequency long-tail distribution.
> - **Q3: Some typos, like: line 16, counteract -> counteracts line 17, enhance > enhances.**
>   - **R3**: Thank you for reading our article carefully and  pointing out these issues, and we will fix them accordingly during the revision phase.
>
> We hope that the above answers will solve your doubts and will continue to pay attention to your follow-up questions.

---

> ### Comment · Reviewer_zwAk · 2023-08-16
>
> Thanks for the author's detailed explanation, some of my concerns are addressed well. Overall, I think the paper is a promising work, so I've slightly increased my scoring to accept.

---

> > ### Author Response · Authors · 2023-08-16
> > **Response to zwAk**
> >
> > Dear Reviewer zwAk:
> >
> > We would like to extend our sincere gratitude to you for the time and effort you have dedicated to reviewing and discussion stage！We greatly appreciate your constructive comments on evaluation of our method in various dimensions, which we believe have significantly contributed to enhancing the quality of our work.
> >
> > Best regards,
> >
> > Paper 12846 authors

---

### Official Review · Reviewer_oe9n · 2023-06-29

**Soundness:** 3 good
**Presentation:** 2 fair
**Contribution:** 3 good
**Rating:** 7
**Confidence:** 3

**Summary:**

This paper tackles the problem of visual models' robustness from the view of frequency components. Inspired by the long-tailed characteristic observed in frequency spectrum, the authors empirically prove that standardly trained models are sensitive to HFC. After analyzing its reasons, the authors proposed a sampling method and verified its effectiveness.

**Strengths:**

1. The authors focuses on a novel perspective of long-tailed distribution in the frequency domain to analyze and improve model performance.
2. The paper provides interesting insights into this problem, revealing the underlying properties of the studied problem.
3. Abundant experiments and analysis are done.
4. The paper is well-written.


**Weaknesses:**

I am not an expert in this area, thus my comments should be treated with caution.
1. Baseline comparisons: BaSS is a new sampling strategy, does there exist any other sampling methods for comparison? PGD-AT is from 2017 and is rather an old method.
2. It would be better to include: the difference in frequency distribution before and after BaSS is adopted.
3. Flaws in writing. E.g. Typo in Table 2 title.

**Questions:**

In Figure 2 second row, why does on Tiny-ImageNet and ImageNet (as opposed to CIFAR10), the tail ranking indices (e.g. "80%" and "Last one") have a high density in the high frequency area? This inconsistency may draw doubts on the generalizability of the conclusions the paper gets.

**Limitations:**

No. The authors should add it.

---

> ### Author Rebuttal · Authors · 2023-08-09
>
> Thanks a lot for your recognition of the novelty of our work！We set out our answers to each of your questions.
>
> - **Q1: Baseline comparisons: BaSS is a new sampling strategy, does there exist any other sampling methods for comparison? PGD-AT is from 2017 and is rather an old method.**
>
>   - **Other methods for comparison?**
>
>     BaSS introduces an innovative sampling strategy for instance-level long-tail challenges, i.e., the long-tail distribution is present within the image frequency domain. Actually, we have not encountered any sampling techniques designed for image features. Prevailing efforts have predominantly concentrated on long-tail scenarios delineated by class or attributes, diverging substantially from our framework centered around the frequency long-tail issues. As a result, a comparative analysis with these methodologies was deemed inappropriate.
>
>   - **Concerns about the PGT-AT that is an old method.**
>
>     We would like to reaffirm our positioning on BaSS, highlighting its broad applicability to a wide range of adversarial training frameworks. We selected two of the most basic adversarial training frameworks, PGT-AT and TRADES. To address your concerns, we expand our experimental scope to include the combination of BaSS with MART. MART is the latest adversarial training framework and is widely used after PGD-AT and TRADES methods. The results are presented in publicly visible PDF material a n d provide compelling evidence that BaSS harmonizes effectively with diverse adversarial training frameworks, resulting in substantial improvements in model robustness and generalization performance.
>
>     It should be noted that we did not choose to combine with the current state-of-the-art (SOTA) algorithms, because the aforementioned algorithms usually face huge time and space overhead, which is a problem that needs to be solved in the process of improving adversarial robustness. The experimental results in Tab 1 and Tab 2 show that our sampling strategy significantly improves the performance of adversarial training model without almost increasing the time and space complexity (details in appendix. Tab 3), and even surpasses the current SOTA algorithms in some scenarios (as shown in Tab 3).
>
> - **Q2: In Figure 2 second row, why does on Tiny-ImageNet and ImageNet (as opposed to CIFAR10), the tail ranking indices (e.g. "80%" and "Last one") have a high density in the high frequency area? This inconsistency may draw doubts on the generalizability of the conclusions the paper gets.**
>
>    - **R2**: We appreciate your careful observation of our experimental results! We also need to point out that on both Tiny-ImageNet and ImageNet, the density peak of the tail ranking index (e.x. last one) is located at the very tail of the spectrum.
>
>      We wish to emphasize that this phenomenon does not impact the consistency of our conclusions drawn from datasets with varying resolutions. Considering the substantial orders of magnitude distinction between the lowest and the highest ranks (illustrated in the first row of Figure 2), we maintain that the model's sensitivity to rankings within this range can be disregarded. Consequently, the perturbations observed in these frequency directions have minimal influence on the model's robustness. Furthermore, it is acknowledged that the ultra-high frequency components in the image often correspond to noise information. This raises the possibility of the model's responsiveness to this particular aspect. Comparable findings can be validated through references [1,2].
>
> - **Other issues:**
>
>   - **It would be better to include: the difference in frequency distribution before and after BaSS is adopted.**
>
>     We visualize the BASS sampled image and the corresponding spectral density distribution curve in Figure 7 of the appendix. Among them, the high-frequency information content of the sampled image is enhanced, and the corresponding spectral energy density is more balanced.
>
>   - **Flaws in writing. E.g. Typo in Table 2 title.**
>
>     We will carefully review the full article and fix these issues during the revision phase.
>
> We will continue to monitor the follow-up questions and discussions of the reviewer, and once again remind that the results of the Q1 experiment are presented in the pdf material shared with all reviewers.
>
>
>
>
>
> **References**
>
> [1] High-frequency component helps explain the generalization of convolutional neural networks, Wang et al.
>
> [2]  Investigating and explaining the frequency bias in image classification, Lin et al.

---

> > ### Comment · Reviewer_oe9n · 2023-08-16
> > **Update**
> >
> > Thank you for your detailed response. The authors have addressed my concerns and I will raise my score to accept.

---

> > > ### Author Response · Authors · 2023-08-16
> > > **Response to oe9n**
> > >
> > > Dear Reviewer oe9n:
> > >
> > >
> > > We once again thank you for your constructive comments and valuable time spent on reviewing and discussion stage!  We are grateful for your meaningful discussion topic and formatting suggestions. They have significantly contributed to enhancing the overall quality of our manuscript.
> > >
> > >
> > >
> > > Best regards,
> > >
> > > Paper 12846 authors

---

### Official Review · Reviewer_P5Z5 · 2023-07-07

**Soundness:** 3 good
**Presentation:** 3 good
**Contribution:** 3 good
**Rating:** 6
**Confidence:** 4

**Summary:**

Using the long-tail features in the spectrum, this paper investigates the model's sensitivity to HFC and discovers patterns of underfitting behavior due to limited information content. The authors propose the Balance Spectrum Sampling (BaSS) strategy, which enhances model learning on HFC. Experimental results show that combining BaSS with existing defense methods improves model performance.

**Strengths:**

1. The paper introduces an interesting perspective by focusing on the long-tailed distribution in the frequency domain and revisits the relationship between high-frequency components (HFC) and model robustness.
2. The under-fitting behavior is explained through the lens of spectral entropy, offering a clear understanding of the low information entropy of HFC.
3. The proposed method achieves improved performance.

**Weaknesses:**

1. The paper lacks a discussion on the existing literature and its relevance to the topic in details, which could provide a stronger foundation for the research. The content in the Introduction section may not be sufficient.
2. More experiments are needed to evaluate the proposed method:
1) The proposed BaSS should be evaluated in combination with more baseline methods. For instance, MART[1] is a method that outperforms TRADES in performance. It would be insightful to see how BaSS performs when incorporated with such stronger baselines.
2) The paper lacks a comparison with other methods, such as AWP[2]. It would provide a more comprehensive evaluation and analysis of the proposed approach.
3. There are some small grammar mistakes in the paper. For instance, in the Abstract, it should be "counteracts and enhances" instead of "counteract and enhance" in lines 16-17. More attention should be given to checking the grammar details throughout the paper.

[1] Improving Adversarial Robustness Requires Revisiting Misclassified Examples. ICLR2020
[2] Adversarial Weight Perturbation Helps Robust Generalization. NeurIPS2020

**Questions:**

1. Is the proposed method also applicable to other visual models, such as Vision Transformer (ViT), in addition to CNNs?

**Limitations:**

No potential negative societal impact

---

> ### Author Rebuttal · Authors · 2023-08-09
>
> We sincerely thank you for affirming the innovativeness of our article and the positive comments about the writing and experiments and we respond to your questions and suggestions below:
>
> - **Q1: The paper lacks a discussion on the existing literature and its relevance to the topic in details, which could provide a stronger foundation for the research. The content in the Introduction section may not be sufficient.**
>
>   - **R1:** Due to the space constraints, the section on related works is presented in the Appendix A. In order to ensure the integrity of the content of the article, we have introduced the core content of the relevant work and the topic in the introduction section. We divided the literatures related to the topic and the corresponding discussions into two parts:
>     - The first section of the related work focuses on the keywords **frequency domain** and **robustness**. In lines 31-40 of the introduction, we presented a discussion on analyzing model robustness in the frequency domain, providing an overview of relevant studies. Additionally, we highlighted two limitations present in existing research, which served as the driving force behind the motivation and problem formulation addressed in our work. A comprehensive description of each individual work is provided in Appendix A.
>
>     - The second part of the related work revolves around the concept of the **long tail**. In lines 48-51 of the introduction, we expounded on the distinct characteristics of the long-tail problem in visual tasks. Building upon this, we drew inspiration from the long-tail distribution observed in the image frequency domain, prompting the formulation of a novel long-tail problem in visual tasks, namely, the instance-wise long-tail problem. Correspondingly, a comprehensive account of the long-tail problem in visual tasks and classical solutions can be found in Appendix A.
>
>     Thank you for your advice, we will carefully consider the location of the references during the revision phase.
>
>
>
>
>
>
>
> - **Q2: Is the proposed method also applicable to other visual models, such as Vision Transformer (ViT), in addition to CNNs?**
>
>   - **R2:** This is a significant and practical question. We endore the importance of considering transformer-based structures (e.x. ViT) and the large-scale dataset (e.x. ImageNet) in data anaylsis,  and we have obtained consistent results regarding the main conclusion of this paper which the robustness issue of vision models mainly stems from the high-frequency components (details is shown in see Appendix D). This indicates that our proposed long-tail scenarios can be well-extended to transformer-based vison models.
>
>     Considering that the ViT model accepts image patches as inputs, wherein each patch itself exists long-tail characteristics in the frequency domain. We believe that the idea of balancing image spectrum energy is also applicable to the improvement of the efficiency of ViT models. In particular, several studies have suggested improving the model's performance by optimizing the utilization of the high-frequency spectrum, which further supports our hypothesis.  However, due to the fact that conducting adversarial training of ImageNet in the ViT model requires substantial computational cost. As a result, we did not perform corresponding experiments in this paper. Nevertheless, we remain committed to exploring and verifying these aspects in our future work.
>
>
>
> - **Q3: More experiments are needed to evaluate the proposed method:**
>
>   - **combination with more baseline methods**
>
>     We supplemented the experiments with MART, as well as the combined applications of experiments on both BaSS and MART. The experiments selected the ResNet18 and WRN34 models and used the CIFAR-10 and CIFAR-100 datasets. We show some of the experimental results below (WRN34, CIFAR-10 over 5 random runs), with detailed results available in a PDF file shared with all authors.
>
>     | Method    | Clean acc  | Attacked by PGD-20 | Attacked by CW | Attacked by AA |
>     | --------- | ---------- | ------------------ | -------------- | -------------- |
>     | MART      | 84.10%     | 57.95%             | 55.53%         | 53.83%         |
>     | MART+BaSS | **89.75%** | **64.43%**         | **63.21%**     | **61.93%**     |
>
>     Our method demonstrates stable performance improvements over superior benchmark methods. We have added MART's results to Table 1 in the main text and included the corresponding references.
>
>   - **comparison with other methods**
>
>     We have compared two types of methods for enhancing robustness in the paper: one is the classical baseline method (which corresponds to a previous question from the reviewers), and the other is the current state-of-the-art (SOTA) method, with SOTA methods referring to the rankings published by Robust Bench. We believe that the author's question corresponds to the comparison of the second type, and following the author's suggestion, we have added Table 3 in the main text to include AWP and other excellent adversarial defense methods, along with comparative analysis and references.
>
>
>
> - **Q4 There are some small grammar mistakes in the paper. For instance, in the Abstract, it should be "counteracts and enhances" instead of "counteract and enhance" in lines 16-17. More attention should be given to checking the grammar details throughout the paper.**
>
>   - **R4:** We sincerely thank you for carefully reviewing our article and identifying these issues for us, and we will fix and try to eliminate similar syntax problems in revision.
>
>
>
> We look forward to following up discussions with you and responding to your follow-up questions, reminding you that detailed  results in Q3 are presented in pdf material shared by all reviewers.

---

> > ### Comment · Reviewer_P5Z5 · 2023-08-20
> > **I have read the response from authors and appreciate their work.**
> >
> > Dear Area Chair and other reviewers
> >
> > I have read the response from authors and the comments from other reviewers. I appreciate the response from authors and think my concerns have been well addressed. Thus, I would like to raise my rating to "weak accept".

---

> > > ### Author Response · Authors · 2023-08-20
> > > **Response to Reviewer P5Z5**
> > >
> > > Dear Reviewer P5Z5:
> > >
> > >
> > >
> > > We sincerely thank you for the valuable time and effort you have dedicated throughout the review and discussion process！ We believe your suggestions regarding experiments and formatting will solidify our work, and the discussion topics you've raised are also immensely valuable for our future exploration. Once again, thank you for acknowledging and appreciating our work.
> > >
> > > Best regards,
> > >
> > > Paper 12846 authors

---

### Official Review · Reviewer_48f5 · 2023-07-09

**Soundness:** 3 good
**Presentation:** 2 fair
**Contribution:** 3 good
**Rating:** 6
**Confidence:** 4

**Summary:**

This paper studies the effect of power imbalance across frequencies in natural image datasets on the accuracy and robustness of neural net classifiers, and empirically argues that the model is underfitted on frequencies with lower power concentration. Then, the paper proposes an augmentation strategy to improve the balance of power distribution per image, and evaluates its proposed method on several datasets and tasks to illustrate its effectiveness in boosting accuracy and robustness to data corruption and adversarial attacks.


**Strengths:**

Studying the effect of spectral power imbalance on the accuracy and robustness of neural networks is a novel and significant effort, and the paper provides several interesting experiments to provide insights into this effect. The proposed augmentation method seems to be effective, and works well in combination with other existing augmentations.

**Weaknesses:**

I think this is a valuable paper overall, however my main concern is a lack of clarity in its explanations and arguments, making several claims that seem incorrect and unjustified to me. Also, the experiments need better organization and more clear and nuanced explanations. I will elaborate my specific concerns and questions below.

1. L115 claims “Inspired by the Pareto principle [34] in long-tailed theory, we propose a stable and quantitative approach”, the paper must be more specific on how the Pareto principle is inspiring the proposed approach and in what sense it is more stable, otherwise such broad statements hinder clarity and readability.

2. Eq 2 set definitions are unclear because r_n is undefined. The explanation given in L117-120 is also unclear since “region” is undefined. Please be more careful and specific when writing definitions that form the foundation of a paper.

3. L133 claims “On one hand, the tail part is often overlooked by the system due to its numerical disadvantage”, what does numerical disadvantage mean? Please be more specific in your statements.

4. In Fig 1 and Section 3.1, the details of how the loss is computed separately for HFC and LFC are missing. Also the colorbar is missing.

5. In Fig 2, top ranks having distributions centered towards the higher frequencies does not mean that the loss is most sensitive to higher frequencies. Please plot Gradient Magnitude versus Frequency Radius to directly observe which frequencies will have the largest effect on loss (largest average magnitude). If lower frequencies receive larger average magnitudes than higher ones, then it cannot be claimed that “perturbations in HFC are more easier to cause errors in the model” as is done in L198.

6. Theorem 1 is just applying the formula plog(p) to the assumed density of p for natural images, this simple application does not need to be a theorem.

7. The paper does not explain or cite why plog(p) is a sensible measure of information on the normalized spectrum (density of p). Note that the normalized spectrum does not have any immediate meaning as a probability distribution, it is a power density function. In particular, the claim in L223 that “This indicates that high-frequency bands in the tail contain substantially less information than low-frequency bands in the head” is misleading, since all that Theorem 1 indicates is that power is concentrated at lower frequencies which is immediate by the assumption on the density of p.

8. Theorem 2 seems immediate from information theory. Could you explain why not simply state that the uniform distribution has maximum Shannon entropy instead of stating Theorem 2?

9. The explanation of the BaSS in section 4.1 is not the same as shown in Algorithm 1 in the Appendix, in that there is no actual sampling happening in Algorithm 1, it is a reweighting of the spectrum based on the ratio of log power density over power density.

10. In the experiments in Tables 1-4, the value of gamma is not specified. It is important to show an ablation study on gamma in all these settings (Fig 3 seems to be doing this, but it is unclear which plot is corresponding to which specific experiment setup from Tables 1-4). Also, the results lack error bars and therefore the significance of the improvements is unclear.

11. Please include ImageNette in Fig 3 results to observe the effect of varying the dataset.

12. In all Tables 1-4, reporting the standalone performance of BaSS is also important. Also, in some situations in Table 4 where BaSS does not help, it is important to comment on why that might be happening.

13. Discussion of related works should be part of the main paper and not in Appendix.

***Some typos***

L1 claims “A widely discussed hypothesis regarding the cause of visual models’ robustness is that they can exploit human-imperceptible high-frequency components”, this seems to be a typo, “models’ lack of robustness” perhaps?

L125: “has not yet to be fully explored” should be “has not been fully explored”.

L136: “regards” be “regarding”.

L321: “worth noted” be “worth noting”.


**Questions:**

See the weaknesses section.

**Limitations:**

See the weaknesses section.

---

> ### Author Rebuttal · Authors · 2023-08-09
>
> Thank you to the reviewers for suggestions that help improve the quality of our work，we make the following responses (e.g., R1 is the response to Q1):
> - **R1:In Line113-117, we have revised it as:**"Previous analysis indicating that a few low-frequency components form the image's main content, while most high-frequency components comprise a minor part. The rule that few factors determine main outcomes is encapsulated by the Pareto Principle and the "80/20" quantitative rule, found across various domains, where 80% of effects are caused by 20% of the reasons. Guided by this rule, we defined the boundary between frequencies using an 80/20 spectral energy ratio." **In Line120, we added the following:**"We experimented with various datasets, finding that different resolution images' high-low frequency boundaries are at a 20% radius from the spectrum's center(see Appendix B.2). This confirms the stability and reasonableness of the classification system using spectral energy and the Pareto Principle."
>
> - **R2: We added the following on eq2**: "$r_n$ is the Euclidean Distance from the nth  frequency band to the center of the spectrum". We give the definition of $r_k$  in L94, $r_n$ and $r_k$ are the differences in footmarks. **We replaced "the region"  with** "the set composed of $\tilde{X}[u,v]$ ".
>
> - **R3:** Considering that the numerical definition will be different in different long tail, **we added the content in L133:** "(e.g., the sample size of the tail class  is far less than that of the head class in the context of class-wise long tail)".
>
> - **R4:**  We have described the details of calculating the loss of high and low frequencies in the Appendix.C. As the complement of  the contour values, we will add colorbar in the revision stage.
>
> - **R5:**  We have implemented the reviewers' recommended visualization method on **Fig1 in public pdf**, and the results show higher gradient amplitude distributions in the HFCs across three datasets, supporting our paper's conclusions. To further address the reviewers' concerns, we consider two points:
>
>   -  Evolutionary trends on five density curves within the top 80%  show specific patterns(e.g., the peak of the density curve on the Tiny-ImageNet dataset exhibits minimal movement). And data points with density values greater than 0 are predominantly situated in the tail regions, indicating the peak will likely not be in the head region.
>   - Instead of solely observing the mean value, which might overlook distribution diversity and abnormalities, we argue that Kernel Density Estimation offers a more accurate representation of the final results.
> - **Response to concerns about theorem in Q6-Q8:**
>     Spectrum of single image is a statistical observation from a probability distribution, reflecting different frequency modes' probabilities. (e.g., high-frequency components are only found in specific locations, such as contours).  Therefore, our definition of probability distribution is based on different frequency domain patterns in frequency domain space as random variables.
>     - Both theorems share the background of representing image distributions in frequency domain and address the long-tail problem in image features instead of a simple calculation formula, crucial in analyzing image data influence on vision model performance.
>     - Theorem 1 innovates in frequency entropy analysis, being superior to conventional image entropy based on grayscale. Its benefits include: (i) Reducing sensitivity to noise; (ii) Compensating for spatial relationship limitations; (iii) Aligning more with human image perceptions.
>     - Theorem 2 combines maximum entropy and theorem 1 for practical value, inspiring the method in Section 4. The results in Section 5 demonstrate its effective application to enhance model robustness-accuracy.
>
> - **R9:**  Image features like spatial pixels and spectral density are not explicitly selectable by the sampling theorem. Thus, inspiring from image sampling in pixel domain, we use the method : (1) Discretizing features with Fourier Transform; (2) Selecting and merging values from various regions, re-weighting different frequency band positions as per Eq.4, and creating a new image through inverse Fourier Transform.
>
> - **R10:**  In Section 4, we outline the purposes and significance of the experiments: (1)Section 4.2 details heuristic experiments to explore the bass strategy, using the gamma parameter for comparing strategies and results.(2)Sections 4.3 and 4.4 present solutions ON two robustness problems (adversarial and corruption) using the bass strategy, with algorithms designed for specific training without gamma-related hyper-parameters. For concerns of error bars, we show the results based on 5 random runs in **Fig1-3 of the public pdf**.
> - **R12**: The individual BaSS results are  shown in Appendix Tables 6-9,  we will reorganize the contents of Table 4 in appendix. For the model's decreasing robustness in the "digital" column of Table 4,  our analysis is:   [1] shows that the digital subcategory  share a common feature: irregular range corruptions from low to high frequencies. AugMix resists them by adopting random data augmentation, enhancing frequency information diversity. The bass method we provided, replacing one AugMix branch,  may affect diversity of frequency domain information in the augmentation data.
>
> - **Other issues**:
>   - (Q11) We have added ImageNette in Fig3 in Appendix.F2.
>   - (Q13) Due to space limitations, please refer to Q1 response onReviewer P5Z5
>   - typos error: We fixed these issues and reviewed the full article.
>
> We follow your valuable advice, clarify the statement and add more experimental results. We sincerely hope that you could reconsider the overall performance and contribution of our articles. We continue to pay attention to any follow-up questions and actively discuss them.
>
> **Reference**
>
> [1]  A fourier perspective on model robustness in computer vision, Yin et al.

---

> > ### Comment · Reviewer_48f5 · 2023-08-10
> > **Thank you for your response**
> >
> > The typo and notation fixes make the paper more clear. Most my concerns are addressed, though I am still not convince about the contribution of theorems, I would appreciate a more direct answer to my questions 6,7,8. Additionally, please make the difference mentioned in my concern 9 very clear in the paper's main body. I'm updating my review accordingly.

---

> > > ### Author Response · Authors · 2023-08-11
> > > **Response to Reviewer 48f5**
> > >
> > > Dear Reviewer 48f5:
> > >
> > >  We would like to express our gratitude to the reviewer for the thorough review and active feedback. Regarding the issues raised in your reply, we engage in further discussions with you. Our responses are as follows:
> > >
> > > - **Concerns about contribution of theorems and direct answer to Q6, Q7, Q8.**
> > >
> > >    - **Response to Q6:**
> > >        - Firstly, Theorem 1 provides a formalized definition of information entropy in the frequency domain, proving that the information content of the head low-frequency components is greater than that of the tail high-frequency components. Previous works, which compared the information sizes of different frequencies starting from image spectrum density, may align with intuitive understanding. However, they don't conform to the mathematical definition of information content (it is also mentioned by the reviewer in q7). Therefore, the hypothesis that low frequencies provide more information for model training and inference was unproven. To the best of our knowledge, no previous work modeled probability distribution in the frequency domain and defined the information content of frequency components. **Hence, our proposed Theorem 1 has made a significant contribution in confirming the proposition about the information content in the frequency domain.**
> > >
> > >        - Secondly, the image information entropy that we defined from a frequency domain perspective has **clear advantages compared to the definition from spatial grayscale values.** These include: (1) Addressing the sensitivity to local noise disturbances; (2) As the frequency domain information comes from global image feature statistics, it can describe advanced image features like textures, whereas grayscale values can only depict brightness information; (3) The results of frequency domain information entropy align more with human perception of different image features.
> > >
> > >        - Lastly, the power-law distribution existing in the image frequency domain is objectively present and widespread. Modeling data distribution in the frequency domain and calculating image information entropy **will be practical for future research work that discusses model performance against backgrounds like information theory.**
> > >     In summary, we consider Theorem 1 to be essential and valuable, possessing the rationality to be recognized as a theorem.
> > >    - **Response to Q7:**  We agree with the idea that each image spectrum can be considered as an observation result of the probabilistic distribution. More precisely, we regard spectral patterns as a type of random variable, with images acting as events composed of various random variables. Statistical analyses of image frequency information can be seen as results of random events, reflecting the probability of different frequency patterns occurring. Intuitively, LFC corresponds to the most common patterns in natural images, typically representing color blocks, whereas HFC only appears at specific locations, such as outlines. Furthermore, the spectral density distribution derived from image datasets reveals a consistency of random variables in the frequency domain following a power-law distribution. This serves as the foundation for our calculation of information entropy across different frequency bands in the dataset.
> > >   - **Response to Q8:** We provide the rationale for introducing Theorem 2 in the description below:
> > >        - Firstly, Theorem 2 is proposed against the backdrop of the long-tail in the frequency domain. Furthermore, the proof of Theorem 2 is based on our definition of the probability distribution in the frequency domain and what was proposed in Theorem 1. Therefore, it's not merely a description based on the maximum entropy principle. Through Theorem 2, we present the ideal spectral distribution which, while founded on the maximum entropy principle, also integrates the random variable $\xi$ in the frequency domain. The ideal distribution is constrained by both $\xi_{l}$ and $\xi_{h}$ in the spectrum.
> > >       - Secondly, Theorem 2 serves as the theoretical foundation for the method we proposed. Considering heuristic design methods that operate on actual image data, merely using the original expression of the maximum entropy principle as guidance might result in a lack of logical context in the paper, as well as neglect of actual variables in the image frequency domain. **Experimental results show that Theorem 2 exhibits universal and effective application benefits in real-world problems.**
> > >
> > > - **Regarding Q9, we have added an detailed explanation on comparing BaSS and other sampling method in Section 4.1 of the paper, based on our previous discussion. We will update this for all reviewers to see in due course.**
> > >
> > > We are grateful for the time and effort the reviewer has invested in reviewing our work, and believe that the suggested changes have enhanced the quality of our research.
> > >
> > > Best regards,
> > >
> > > Paper12846 Authors

---

### Official Review · Reviewer_fSNP · 2023-07-10

**Soundness:** 3 good
**Presentation:** 2 fair
**Contribution:** 2 fair
**Rating:** 5
**Confidence:** 4

**Summary:**

This paper addresses the problem of model robustness, which is of great importance in the machine learning field. The authors revisit this problem and discuss the relationship between high-frequency components and model robustness. They verify the effectiveness of the proposed method on several datasets compared to other baselines.

**Strengths:**

This paper addresses the model robustness problem, which is of great importance to the machine learning field.

The authors verify the effectiveness of the proposed method on several datasets compared to other baselines.

The proposed method is theoretically sound with supporting proofs.


**Weaknesses:**

How many times were these experiments conducted, and is it necessary to report the mean and standard variance of the results?

I noticed that the current experiments were mainly conducted on toy datasets, such as CIFAR or ImageNet subsets. What about scaling up the experiments to large-scale datasets in the real world?

The current experiments mainly focus on the PGD attack. Have you considered experimenting with other types of attacks to evaluate the performance of the proposed method in different scenarios? Additionally, apart from the single baseline PGD-AT, have you tested other baselines? Is the improvement consistent for all variants?

Although the proposed method is supported by theoretical proofs, it seems relatively naive. Further robustness analysis is needed for some hyperparameters in Eq. 4.


**Questions:**

(Just a remark) The figures in the supplementary materials are intuitive, and reorganizing the paper could improve its readability.

---

> ### Author Rebuttal · Authors · 2023-08-09
>
> We are grateful to the reviewer for your time and effort in evaluating our manuscript. We have responded to each question in detail below：
> - **Q1:Have you considered experimenting with other types of attacks to evaluate the performance of the proposed method in different scenarios? Additionally, apart from the single baseline PGD-AT, have you tested other baselines? Is the improvement consistent for all variants?**
>   - **Rich attack scenarios:** We have evaluated on two types of model robustness that are of primary concern in the community: adversarial robustness and corruption robustness. For evaluations of adversarial robustness, we used  the efficient attack algorithms: PGD, CW, and AutoAttack(AA). The results are presented in Tab1-3 in the paper, and Tab2 in the appendix. For evaluations of corruption robustness, we selected 15 pixel-level image corruption algorithms and set five intensity levels under each perturbation algorithm, summarizing all the algorithms into four categories as shown in Tab4. The results in Tab1-4 show that our proposed method can effectively defend against all the above attack scenarios.
>   - **Adding more baselines:** In the paper, we tested two adversarial training baselines: PGD-AT and TRADES. We report results for **TRADES on the ImageNette below**. In addition, we have report  results on the latest adversarial training baseline, MART in **Fig3 in public pdf**.
>     | Method     | Clean acc | Attacked by PGD-10 | Attacked by AA |
>     | ---------- | --------- | ------------------ | -------------- |
>     | TRADE      | 64.79%    | 51.71%             | 35.50%         |
>     | TRADE+BaSS | 66.90%    | 59.96%             | 42.80%         |
>
>     The above results demonstrate that BaSS method can be applied to enhance the robustness of various  baseline algorithms and ensures stability under various attack algorithm tests.
> - **Q2:  What about scaling up the experiments to large-scale datasets in the real world?**
>   - **R2:** Conducting experiments on adversarial defenses in large-scale datasets is not an easy task. Due to issues with computational resources and time, we were unable to provide you with relevant experimental results during the rebuttal phase. By referring to related work in the same field[1-2], our experimental results on the ImageNet subset already possess convincing evidence for evaluation on large datasets. Additionally, to address your concern, we summarize the advantages of BaSS, which can support training on even larger-scale datasets.
>     - **Low Time-Space Complexity.** BaSS  can be regarded as a pre-processing operation on images. Therefore, it will not incur any additional overhead in memory resources; Moreover, the results in Appendix Tab3 indicate that bass almost does not increase the time overhead during the training process.
>     - **Hyperparameter-Free.** For the hyperparameters of the training process, the algorithm with BaSS adopts the same settings as the baseline. Therefore, it does not have to face issues such as hyperparameter optimization. As a result, even training on larger datasets will not increase the difficulty of optimization.
>     - **Furthermore**, model robustness is widely considered to be sensitive to the resolution of the training images and the number of categories in the dataset. In our experiments, we set up comparative experiments for the above factors. The experimental results show that BaSS can effectively enhance model robustness across datasets with multiple different resolutions and different numbers of categories.
> - **Q3: How many times were these experiments conducted, and is it necessary to report the mean and standard variance of the results?**
>   - **R3:** Thank for your question and suggestion, our experimental results are based on  3 random runs, the standard deviations of 3 runs are very small, which hardly effect the results. In the **fig1 and fig2 of public pdf**, we reported the results and the standard deviations of 5 runs, which are still very stable.
> - **Q4: Although the proposed method is supported by theoretical proofs, it seems relatively naive. Further robustness analysis is needed for some hyperparameters in Eq. 4.**
>   - **R4:** To address the concern of the reviewers, we would like to clarify the motivation behind the theorem  and conclude its importance.
>      - Both theorems share the background of representing image distributions in frequency domain and address the long-tail problem in image features instead of a simple calculation formula, crucial in analyzing image data influence on vision model performance.
>      - Theorem 1 innovates in frequency entropy analysis, being superior to conventional image entropy based on grayscale. Its benefits include: (i) Reducing sensitivity to noise; (ii) Compensating for spatial relationship limitations; (iii) Aligning more with human image perceptions.
>      - Theorem 2 combines maximum entropy and theorem 1 for practical value, inspiring the method in Section 4. The results in Section 5 demonstrate its effective application to enhance model robustness-accuracy.
>     In Eq. 4, where the parameter *B* is determined by the size of  image, and parameter *τ* is the constant *e*, we will provide a clearer explanation during the revision stage. We do not wish to introduce hyperparameter settings to the method, thereby increasing its training difficulty across different datasets. From another perspective, the adjustment of parameters might lead to better results, which could become a direction for optimizing BaSS in our future work.
>
> We will continue to follow and respond to any questions regarding the content of the article following the review. Meanwhile, we sincerely hope that the reviewer could reassess the contributions of our work.
>
> **References**
> [1]  Efficient and effective augmentation strategy for adversarial training. NeurIPS 2022
> [2] Robust Learning Meets Generative Models: Can Proxy Distributions Improve Adversarial Robustness? .ICLR 2022

---

> > ### Comment · Reviewer_fSNP · 2023-08-11
> > **Response**
> >
> > I thank the authors for the detailed response, which addresses most of my concerns. I am updating my rating to borderline accept.

---

> > > ### Author Response · Authors · 2023-08-11
> > > **Response to Reviewer fSNP**
> > >
> > > Dear Reviewer fSNP:
> > >
> > > Thank you for your suggestion to expand on the evaluation of  our method. This has indeed strengthened our paper.  We sincerely appreciate the effort you put into the entire review process !
> > >
> > > Best regards,
> > >
> > > Paper12846 authors

---

### Author Rebuttal · Authors · 2023-08-10

Dear Area Chair and Reviewers,

We would like to thank all reviewers for their thoughtful comments and insights that greatly helped improve our work. We have carefully reviewed and responded to every question from all reviewers, providing a clearer explanation of the content of the article in question, and adding more experimental results to validate our conclusions and discussions. We have summarized the following modifications and additions:

- Added more experiments

  - We supplemented the experiments with MART and MART+BaSS on the CIFAR10 and CIFAR100 datasets, and the results are shown in Tab1-3 of the PDF file.
  - We added experiments with TRADES and TRADES+BaSS on the large-scale dataset ImageNette, and the results are shown in the response to reviewer fSNP.
  - We supplemented the visualization results of fig1 in paper to further validate our conclusions, and the results are shown in fig1 of the PDF.

  Due to computational resource constraints, only partial experimental results are shown, and we will supplement the complete experimental tables during the discussion and revision stage.

- Supplement to the main text and appendix,

  - We have improved some statements in Sections 2-3 of the article, which can be seen in the response to the reviewer 48f5.
  - We added the results of fig2 on imagenette in appendix F.2.
  - We added content and corresponding experimental results of discussions with reviewer zwAk in appendix F.2.
  - We added a table contrasting the individual results of BaSS, according to reviewer 48f5 Q12.

- We have corrected all the typos issues raised by the reviewers.

During the discussion stage, we will continue to pay attention to the reviewers' follow-up questions and suggestions, and supplement the above content.

---

> ### Author Response · Authors · 2023-08-11
>
> Dear Area Chair and Reviewers,
>
> Thank you for the positive responses received so far from reviewer 48f5 and fSNP. Additionally, we have incorporated suggestions from reviewer 48f5 and added content that we believe will enhance the clarity of our method's description:
>
> - Supplement to the main text
>    - In Section 4.1, based on our response to reviewer 48f5's Q9 feedback, we have added a description detailing the differences between our proposed BaSS and other sampling algorithms.
>
> We will continuously share the updates in a public area visible to all reviewers. We are looking forward to responses from other reviewers.
>
>
>
>
> Best regards,
>
> Paper12846 authors

---

### Decision · Program_Chairs · 2023-09-21

**Decision:**

Accept (poster)

**Comment:**

This paper was reviewed by five experts in the field. Based on the reviewers' feedback, the decision is to recommend the paper for acceptance to NeurIPS 2023. The reviewers did raise some valuable concerns that should be addressed in the final camera-ready version of the paper. The authors are encouraged to make the necessary changes to the best of their ability. We congratulate the authors on the acceptance of their paper!